# A Multi-Class Intrusion Detection System for DDoS Attacks in IoT Networks Using Deep Learning and Transformers

**DOI:** 10.3390/s25154845

**Published:** 2025-08-06

**Authors:** Sheikh Abdul Wahab, Saira Sultana, Noshina Tariq, Maleeha Mujahid, Javed Ali Khan, Alexios Mylonas

**Affiliations:** 1Department of Computing and Technology, H-9 Campus, Iqra University, Islamabad 44000, Pakistan; 232886@students.au.edu.pk (S.A.W.); saira.sultana@iqraisb.edu.pk (S.S.); maleeha.mujahid@iqraisb.edu.pk (M.M.); 2Department of Avionics Engineering, Main Campus PAF Complex E-9, Air University, Islamabad 44000, Pakistan; 3Department of Artificial Intelligence and Data Science, National University of Computer and Emerging Sciences, Islamabad 44000, Pakistan; 4Department of Computer Science, Cybersecurity and Computing Systems Research Group, University of Hertfordshire, Hertfordshire AL10 9AB, UK; j.a.khan@herts.ac.uk

**Keywords:** Internet of Things security, Distributed Denial of Service, Intrusion Detection System, Deep Learning, Convolutional Neural Network, Transformer, Synthetic Minority Over-sampling Technique, anomaly detection

## Abstract

The rapid proliferation of Internet of Things (IoT) devices has significantly increased vulnerability to Distributed Denial of Service (DDoS) attacks, which can severely disrupt network operations. DDoS attacks in IoT networks disrupt communication and compromise service availability, causing severe operational and economic losses. In this paper, we present a Deep Learning (DL)-based Intrusion Detection System (IDS) tailored for IoT environments. Our system employs three architectures—Convolutional Neural Networks (CNNs), Deep Neural Networks (DNNs), and Transformer-based models—to perform binary, three-class, and 12-class classification tasks on the CiC IoT 2023 dataset. Data preprocessing includes log normalization to stabilize feature distributions and SMOTE-based oversampling to mitigate class imbalance. Experiments on the CIC-IoT 2023 dataset show that, in the binary classification task, the DNN achieved 99.2% accuracy, the CNN 99.0%, and the Transformer 98.8%. In three-class classification (benign, DDoS, and non-DDoS), all models attained near-perfect performance (approximately 99.9–100%). In the 12-class scenario (benign plus 12 attack types), the DNN, CNN, and Transformer reached 93.0%, 92.7%, and 92.5% accuracy, respectively. The high precision, recall, and ROC-AUC values corroborate the efficacy and generalizability of our approach for IoT DDoS detection. Comparative analysis indicates that our proposed IDS outperforms state-of-the-art methods in terms of detection accuracy and efficiency. These results underscore the potential of integrating advanced DL models into IDS frameworks, thereby providing a scalable and effective solution to secure IoT networks against evolving DDoS threats. Future work will explore further enhancements, including the use of deeper Transformer architectures and cross-dataset validation, to ensure robustness in real-world deployments.

## 1. Introduction

The Internet of Things (IoT) has enabled various industries to provide real-time surveillance, automation, and improved decision-making capabilities. For example, healthcare, smart homes, smart cities, and even industrial systems have adopted IoT technologies, considerably enhancing their efficiency and productivity [1,2]. However, the same technologies enabling increased connectivity also heighten the cybersecurity risks, especially for IoT systems. IoT systems are significantly at risk of Distributed Denial of Service (DDoS) attacks, which can turn off devices and networks by flooding them with malicious traffic [3,4]. The inherently low computational power, restricted storage, and limited energy resources make IoT devices easier targets for such DDoS attacks and challenge the implementation of efficient, robust, and enduring security measures [5].

Different Intrusion Detection Systems (IDS) have relied on fundamental Machine Learning (ML) as their primary technique in response to these challenges [6,7,8]. It has proven to be effective to an extent. However, there are still glaring issues that these systems fail to solve. Classical ML approaches are known to not perform well with data with many variables, limiting generalization and, therefore, achievement of target detection [9,10,11]. When looking at the context of the problem, the classical model performs poorly since it struggles to mediate the divide between sophisticated DDoS traffic and benign entities. Another aspect to take into consideration is the lack of automation and feature manipulation, which leads to the need for manual intervention. Because of this, the model demonstrates gaps and limitations against new threats [12,13]. The model also struggles in dealing with unknown attack variants. It is also important to highlight that classical models bound these problems with imbalanced data sets, a common characteristic of intrusion detection leading to biased outcomes in terms of detection results.

In recent years, Deep Learning (DL) models have surfaced as this growing alternative for solving these issues and have the potential to improve IDS capabilities significantly [6,14,15,16]. Algorithms like Convolutional Neural Networks (CNNs), Deep Neural Networks (DNNs), and, more recently, Transformers do particularly well at providing accuracy and generalization with high-dimensional data because they automatically extract intricate patterns from large datasets [17,18]. However, the existing solutions for DL-based IDS will still have significant gaps, such as poor computational resource efficiency, over-reliance on optical training, and poor data balance management [19]. Also, the limited computing resources such as processing, memory, and energy available in IoT environments usually make the application of standard DL models impractical [20].

This work addresses the above gaps and develops an efficient DL framework for DDoS attack detection in IoT networks with limited resources. The proposed approach incorporates robust class imbalance mitigation through normalization, scaling, and SMOTE-enhanced data preprocessing. This work presents specialized architectures based on CNNs, DNNs, and Transformers customized to the low-resource constraints posed by IoT devices. The models use low-power hardware, lower requirements for computation, and improved control of overfitting/underfitting to increase generalization and decrease overfitting. The innovative integration of all these factors will increase the performance of the models when used in severely resource-constrained IoT environments.

The key contributions of this paper are summarized as follows:Development of optimized CNN, DNN, and Transformer architectures uniquely adapted for the effective detection of DDoS attacks in resource-constrained IoT environments, emphasizing lightweight computation and rapid response capability.Implementation of advanced preprocessing techniques, including Min–Max normalization and SMOTE, to effectively mitigate class imbalance, thereby significantly improving the accuracy and reliability of intrusion detection.Demonstration of a realistic deployment scenario in an IoT-based smart environment to validate the practical applicability of the proposed detection models in operational network settings.Utilizing the latest CIC-IoT-2023 dataset, which is specifically designed for IoT environments for binary, 3-, and 12-class classification tasks.

The rest of this paper follows with a review of related work in Section 2. Section 3 and Section 4 detail a use case related to DDoS attacks in IoT enviornments and the proposed methodology, respectively. After that, the experimental setup and training process are described in Section 5. The results and performance metrics are discussed in Section 6. It concludes with a discussion and final remarks in Section 7.

## 2. Related Work

The IoT has grown rapidly in recent years. It has introduced new areas of concern, such as security, particularly about DDoS attacks. Recent studies have aimed towards developing effective DL mitigation strategies using the CIC-IoT 2023 dataset. This dataset has recently grown in popularity among scholars analyzing network traffic due to its rich sequential information and the attention mechanisms of Transformer architectures. Notably, applying DL techniques has provided ideal results, but challenges still exist regarding real-time detection, explainability, and resource utilization.

Tseng et al. [21] employed Transformers on the CIC-IoT 2023 dataset, achieving high accuracy, though their work lacks robustness testing against adversarial attack strategies. Similarly, Wasswa et al. [22] explored Vision Transformers (ViTs) for botnet classification, but their narrow focus excluded broader DDoS attack vectors, limiting generalizability. Hybrid DL models have also gained traction. Mahdi et al. [23] combined CNNs with LSTMs to improve detection accuracy; however, the computational demands of their model hinder its viability in real-time, resource-constrained IoT systems. Lamba et al. [24] compared AI-driven IDS models and confirmed Transformer superiority, yet overlooked class imbalance issues, which can distort multiclass performance evaluations.

Efforts to integrate explainability and generalization have also emerged. Ullah et al. [25] proposed a multimodal CNN-BERT architecture but relied heavily on manual feature engineering, affecting adaptability. Baral et al. [26] applied XAI principles in an end-to-end framework, enhancing transparency, though throughput limitations remain. In parallel, Li et al. [27] introduced LLM-driven IDS agents, offering interpretability but lacking empirical validation for scalability. To mitigate data scarcity, Almaraz-Rivera et al. [28] investigated self-supervised learning, which improved generalization but remained dependent on labeled fine-tuning data. Hizal et al. [29] implemented a two-stage binary-multiclass DL IDS with efficient feature pruning; however, their reliance on static features risks overlooking emerging attack vectors.

Alternative strategies like GANs and federated learning have also contributed to this domain. Sharma et al. [30] enhanced attack detection using GAN-augmented ML models, while Bertoli et al. [31] facilitated decentralized detection with privacy-preserving federated setups. Yao et al. [32] utilized BiGANs to uncover novel threats, though their approach was computationally heavy. Thiyam and Dey [33] addressed class imbalance via SMOTE and TOMEK-Link, stabilizing minority class detection. Other notable methods include Soft-Ordering CNNs [34], which combined hierarchical modeling with outlier detection for anomaly identification, and hybrid CNN–BiLSTM approaches [35], which captured spatial and temporal traffic features more effectively. Zhao et al. [36] applied attention-based FlowTransformer models with high accuracy, though scalability remained a concern.

In contrast, Karimy et al. [37] deployed a lightweight 1D-CNN for real-time anomaly detection on edge devices, showcasing the feasibility of deep models in constrained IoT environments. Recent work by Bolat et al. [38] proposed a software-defined intrusion detection framework for IoT edge networks, leveraging SDN and edge computing to detect DDoS attacks. While effective in architectural design, the approach does not integrate deep learning or address multiclass classification challenges. Similarly, Liu et al. [39] introduced MalDetect for encrypted malware traffic detection using early packet features and online learning via Random Forest. Though efficient in detecting TLS-encrypted flows, their method lacks adaptation to IoT-specific threats and does not incorporate deep neural architectures. These studies emphasize performance and early detection but leave open challenges in scalability, deep learning integration, and real-time adaptability, which our work addresses, see Table 1.

The reviewed literature highlights the continuous evolution of DL techniques for intrusion detection in IoT environments. While Transformer-based models have shown promising results in handling large-scale network traffic data, their high computational requirements remain a challenge for real-time applications. Future research should focus on lightweight and efficient IDS frameworks that can operate seamlessly in resource-constrained IoT environments while maintaining high detection accuracy.

Despite the advancements in DL and Transformers for DDoS detection, several limitations persist. First, many studies do not account for real-time deployment challenges, including latency and energy constraints in IoT networks. Second, adversarial robustness remains a critical concern, as attackers continuously evolve their strategies to evade detection systems. Finally, while explainability is an emerging focus, existing methods often sacrifice detection accuracy for interpretability, necessitating a trade-off between performance and transparency.

Future research should address these gaps by integrating lightweight Transformer models optimized for edge computing environments. Additionally, the development of adversarially robust detection mechanisms can enhance the resilience of DL-based security systems. Lastly, hybrid approaches that combine rule-based heuristics with data-driven learning can improve both interpretability and detection accuracy, leading to more effective IoT security solutions.

## 3. Use Case: DDoS Attacks in IoT Environments

In recent years, the rapid proliferation of IoT devices across critical sectors such as healthcare, smart homes, and industrial automation has led to an unprecedented level of network connectivity and interdependency. These interconnected IoT devices facilitate significant operational efficiency, real-time monitoring, and enhanced decision-making capabilities. However, this widespread connectivity introduces substantial cybersecurity vulnerabilities, particularly susceptibility to DDoS attacks.

In a typical DDoS scenario in an IoT environment, malicious actors compromise multiple IoT devices—often due to inherent security flaws or weak authentication protocols—to create botnets. These compromised devices collectively launch coordinated attacks by flooding targeted nodes or servers with overwhelming volumes of network traffic, effectively exhausting available resources and rendering the services unavailable. The consequences of such attacks may be catastrophic, such as prolonged service outages, patient safety risks in healthcare systems, disrupted smart city operations, and moderate-to-severe financial damage in the industrial domain.

An example is a smart healthcare IoT system consisting of medical sensors that monitor a patient’s health and send critical data for real-time monitoring. A DDoS attack may interfere, resulting in delays in emergency action systems and death. The same applies to industrial IoT systems, where DDoS attacks can be targeted to stop automated processes, resulting in increased downtime, lower productivity, and significant economic losses.

Because of these consequences, there is an emerging need for efficient techniques to deal with DDoS attacks designed for IoT networks with specific limitations such as low power, limited processing capability, memory, and any other resource that may be constrained. Therefore, developing a lightweight, efficient, and reliable IDS capable of rapidly identifying and mitigating DDoS attacks is essential for maintaining the integrity, availability, and resilience of IoT-enabled critical infrastructures.

## 4. Proposed Methodology

This section describes the systematic methodology employed in the study, detailing each step comprehensively. The methodology is structured into two main phases: (1) Data Preprocessing and (2) Model Development and Training. Each phase includes explicit subsections that clarify the mathematical formulations, algorithmic processes, and rationale behind the methods used. Table 2 represents the list of symbols and notations used in this section. In addition, Figure 1 illustrates the proposed methodology at a glance.

### 4.1. Dataset Description

The dataset utilized in this study is the CICIoT2023 dataset (Available online: https://www.unb.ca/cic/datasets/iotdataset-2023.html (accessed on 15 April 2025)), a publicly available dataset specifically designed for IoT security research. This dataset consists of network traffic data collected from 105 IoT devices, covering a diverse range of attacks, including DDoS, Denial of Service (DoS), Reconnaissance, Web-Based, Brute Force, Spoofing, and Mirai attacks. The dataset contains a total of 46,686,579 samples, distributed across 33 different attack types.

Table 3 provides an overview of the dataset, including the number of samples per category, attack diversity, and distribution.

### 4.2. Data Preprocessing

The preprocessing phase includes multiple steps to clean, normalize, and balance the dataset, ensuring optimal performance for DL models. The following preprocessing steps were applied:Feature Selection: The original dataset contained 46 features. Features with zero variance or missing values were removed, retaining 37 meaningful features for training.Class Balancing with SMOTE: Due to significant class imbalance, the Synthetic Minority Over-sampling Technique (SMOTE) was applied to generate synthetic samples for minority classes, ensuring a balanced dataset. SMOTE was mathematically implemented using Equation  (Equation 1).(1)$$x_{new}=x_{i}+\alpha \left(x_{j}-x_{i}\right)$$
where $x_{i}$ is a minority class instance, $x_{j}$ is its nearest neighbor, and α is a random value in [0,1]. It is to be noted that the SMOTE algorithm was applied solely to the training set in each fold of cross-validation. The test set in every case remained untouched, ensuring that synthetic samples did not influence evaluation and data leakage was strictly avoided. In the 2-class and 3-class tasks, the minority “Attack” and “Other” classes were oversampled, respectively. For the 12-class setting, underrepresented classes such as Class 6 (Infiltration) and Class 7 (Heartbleed) were synthetically balanced to ensure fair learning across all categories.Normalization: Min–Max normalization was applied to ensure uniform feature scaling, defined as Equation  (Equation 2).(2)$$x^{\prime }=\frac{x-x_{min}}{x_{max}-x_{min}}$$
where $x_{min}$ and $x_{max}$ represent the minimum and maximum values of the feature.Random Subset Selection: A subset of 3,000,000 samples was randomly selected from the dataset for efficient training, while maintaining class balance.Dataset Partitioning: The dataset was partitioned into three classification tasks:Binary Classification: Benign vs. Attack (DDoS and Non-DDoS combined).Three-Class Classification: Benign, DDoS, and Non-DDoS.Multi-class Classification: Benign and 12 attack types (10 DDoS subclasses and 1 Non-DDoS).Duplicate Removal: Duplicate records were identified and removed to prevent model overfitting.Log Normalization: Features with a wide range of values were log-transformed to stabilize variance, as shown in Equation (Equation 3).(3)$$x^{\prime }=\log\left(1+x\right)$$

Table 4 summarizes the distribution and overview of dataset samples.

With these preprocessing steps, the dataset was prepared for training DL models with balanced class distribution and optimized feature representations.

### 4.3. CNN for IoT DDoS Attack Detection

A CNN is an effective DL model that extracts spatial features from structured data. CNNs are widely used in network intrusion detection due to their capability to automatically learn hierarchical patterns from network traffic data. The use of convolutional layers helps in detecting local dependencies, while pooling layers enable dimensionality reduction, making CNNs computationally efficient.

For DDoS attack detection in IoT networks, CNNs process network traffic flows, identifying anomalous behavior that distinguishes normal and malicious activities. Unlike traditional ML models, CNNs extract features directly from raw input data, eliminating the need for manual feature engineering. This capability is crucial in IoT environments where traffic data is highly dynamic and complex. The components of the proposed CNN model are as follows:Convolutional Layer: A convolutional layer filters the input features to extract relevant spatial information. The processes that occur inside a convolutional layer are described in Equation (Equation 4).(4)$$Z_{i,j}^{\left(l\right)}=\sum_{p=1}^{P}\sum_{q=1}^{Q}W_{p,q}^{\left(l\right)}\cdot A_{i+p,j+q}^{\left(l-1\right)}+b^{\left(l\right)}$$
where $Z^{\left(l\right)}$ represents the output feature map at layer *l*, $W^{\left(l\right)}$ denotes the kernel weights of size P×Q, $A^{\left(l-1\right)}$ is the activation from the previous layer, and $b^{\left(l\right)}$ is the bias term. This operation enables CNNs to detect attack patterns in network traffic, identifying localized anomalies in packet sequences.Batch Normalization: By performing normalization on the mini-batch on which training is being executed, batch normalization speeds up convergence and stabilizes the training. It can be stated as Equation (Equation 5).(5)$${\hat{x}}^{\left(l\right)}=\frac{x^{\left(l\right)}-\mu^{\left(l\right)}}{\sqrt{\sigma^{2(l)}+\epsilon }}$$
where $\mu^{\left(l\right)}$ and $\sigma^{2(l)}$ are the mean and variance for the batch, respectively, and ϵ is a small constant for numerical stability. Normalizing the activations as described allows the networks trained to be more effective and more resilient to various IoT traffic patterns.Spatial Dropout for Regularization: To prevent overfitting, dropout is commonly used in which neurons are randomly turned off during training. In CNNs, SpatialDropout1D is utilized. It drops entire feature maps instead of individual activations, which increases generalization for sequence-based data, as shown in Equation (Equation 6).(6)$$A^{\left(l\right)}=\mathrm{Dropout}\left(A^{\left(l\right)},p\right)$$
where *p* is the probability of dropout; it also makes sure that the CNN model does not overfit on specific patterns, making it resilient to adversarial changes in DDoS attack traffic.Global Average Pooling (GAP): It offers an approach for reducing the dimensionality of the feature maps while retaining the spatial relationship, as shown in Equation (Equation 7).(7)$$GAP_{i}=\frac{1}{n}\sum_{j=1}^{n}Z_{ij}$$
where *n* represents the number of spatial elements. GAP enhances model interpretability by assigning a single value per feature map, reducing overfitting in small datasets like IoT attack logs.Fully Connected Layer: The fully connected layer maps extracted features to attack classes using the softmax activation function, ensuring probabilistic classification, as shown in Equation (Equation 8).(8)$$P\left(y=k|X\right)=\frac{e^{z_{k}}}{\sum_{j=1}^{C}e^{z_{j}}}$$
where *C* is the number of classes. This layer produces final predictions, assigning attack labels to network traffic.Loss Function and Optimization: The CNN model is trained using the sparse categorical cross-entropy loss, which is defined as Equation (Equation 9).(9)$$L=-\sum_{i=1}^{N}y_{i}\log\left({\hat{y}}_{i}\right)$$
where $y_{i}$ and ${\hat{y}}_{i}$ denote the true labels and predicted probabilities. The optimization follows an exponential decay learning rate to enhance model stability, as shown in Equation (Equation 10).(10)$$\eta_{t}=\eta_{0}\times {\mathrm{decay}\_\mathrm{rate}}^{\left(\frac{\mathrm{step}}{\mathrm{decay}\;\mathrm{steps}}\right)}$$This strategy helps adapt learning rates dynamically, preventing stagnation in optimization.

The training and evaluation process for CNN-based DDoS attack detection follows structured steps. Algorithm 1 describes the CNN training and evaluation process for IoT DDoS attack detection. It begins with data normalization and class balancing before training the CNN. The convolutional layers extract spatial attack patterns, while dropout and batch normalization improve generalization. The network iteratively updates its weights based on cross-entropy loss, and the learning rate is adjusted dynamically. The trained model is validated using standard evaluation metrics.
**Algorithm 1** CNN-based IoT attack detection**Require:**
Dataset D={X,y}, learning rate $\eta_{0}$, batch size *B*, epochs *E*, folds *K***Ensure:**
Trained model M, results R  1: Normalize features: $x^{\prime }=\frac{x-x_{\min}}{x_{\max}-x_{\min}},\forall x\in X$  2: Initialize model weights $W,b∼N(0,\sigma^{2})$  3: Initialize result set R←∅  4: F← Stratified *K*-fold split on (X,y)  5: **for all **
$(X_{\mathrm{train}},y_{\mathrm{train}},X_{\mathrm{test}},y_{\mathrm{test}})\in F$
**do**  6:     $\left(X_{\mathrm{train}}^{\mathrm{res}},y_{\mathrm{train}}^{\mathrm{res}}\right)\leftarrow \mathrm{SMOTE}\left(X_{\mathrm{train}},y_{\mathrm{train}}\right)$  7:     **for** e=1 to *E* **do**  8:         Shuffle$\left(X_{\mathrm{train}}^{\mathrm{res}},y_{\mathrm{train}}^{\mathrm{res}}\right)$  9:         **for all** $B_{i}=\left(X_{B},y_{B}\right)\subset \left(X_{\mathrm{train}}^{\mathrm{res}},y_{\mathrm{train}}^{\mathrm{res}}\right)$ **do**  10:            $Z=W*X_{B}+b$  11:            $\hat{Z}=\frac{Z-\mu }{\sqrt{\sigma^{2}+\epsilon }}$  12:            $\tilde{Z}=\mathrm{Dropout}\left(\hat{Z},p\right)$  13:            $\hat{y}=\frac{e^{\tilde{Z}}}{\sum e^{\tilde{Z}}}$  14:            $L=-\sum_{i=1}^{N}y_{i}\log\left({\hat{y}}_{i}\right)$  15:            $W\leftarrow W-\eta \frac{\partial L}{\partial W},\;b\leftarrow b-\eta \frac{\partial L}{\partial b}$  16:         **end for**  17:         $\eta \leftarrow \eta_{0}\times \lambda^{\frac{e}{E}}$  18:         $R_{e}\leftarrow \mathrm{Evaluate}\left(M,X_{\mathrm{test}},y_{\mathrm{test}}\right)$  19:         **if** EarlyStopping($R_{e}$) **then**  20:              Break  21:         **end if**  22:     **end for**  23:     Append $R_{e}$ to R  24: **end for**  25: **return**
M,R

### 4.4. DNN for IoT DDoS Attack Detection

A DNN is a multi-layered feed-forward architecture capable of capturing complex, nonlinear relationships in high-dimensional data. Unlike convolutional networks, which focus on spatial patterns, DNNs excel in learning global dependencies across features, making them suitable for IoT traffic analysis where attacks are often hidden in multi-feature interactions. For DDoS attack detection in IoT environments, DNNs classify network traffic into normal and attack classes by leveraging deep feature transformations. The network’s hierarchical structure enables it to detect subtle deviations indicative of cyber threats. The proposed DNN model efficiently processes IoT traffic logs, improving detection accuracy and robustness. The major components of a DNN model are discussed bellow:Input Layer and Feature Representation: The input to the DNN is a set of normalized feature vectors extracted from network traffic flows, as shown in Equation (Equation 11).(11)$$X=\left[x_{1},x_{2},...,x_{n}\right]\in R^{m\times n}$$
where *m* is the number of samples, and *n* is the number of extracted features per sample. Each input feature captures statistical and time-based properties of IoT traffic, forming a structured multi-dimensional feature space.Fully Connected Layers: DNNs use fully connected layers (dense layers) to transform inputs through a sequence of weighted linear and nonlinear transformations, as shown in Equation (Equation 12).(12)$$Z^{\left(l\right)}=W^{\left(l\right)}A^{\left(l-1\right)}+b^{\left(l\right)}$$
where $Z^{\left(l\right)}$ represents the pre-activation output of layer *l*, $W^{\left(l\right)}$ is the weight matrix of the layer, and $b^{\left(l\right)}$ is the bias vector. Each fully connected layer enables the DNN to learn relationships between IoT traffic attributes and classify attack types.Activation Function (ReLU): To introduce nonlinearity, the Rectified Linear Unit (ReLU) activation function is applied using Equation (Equation 13).(13)$$A^{\left(l\right)}=\max\left(0,Z^{\left(l\right)}\right)$$ReLU accelerates training by avoiding vanishing gradient issues, ensuring efficient convergence in large-scale IoT datasets.Dropout for Regularization: To prevent overfitting, dropout regularization is applied, where neurons are randomly deactivated during training, as shown in Equation (Equation 14).(14)$$A^{\prime (l)}=\mathrm{Dropout}\left(A^{\left(l\right)},p\right)$$
where *p* is the dropout probability.Dropout improves generalization performance, ensuring that the DNN does not memorize training patterns but instead learns intrinsic attack behaviors.Output Layer and Softmax Classification: The final layer maps deep features to output classes using the softmax activation function, as shown in Equation (Equation 15).(15)$$P\left(y=k|X\right)=\frac{e^{z_{k}}}{\sum_{j=1}^{C}e^{z_{j}}}$$
where *C* is the number of attack classes. This transformation converts raw logits into probabilistic outputs, ensuring that predictions are interpretable and confidence-based.Loss Function and Optimization: Training is performed using the sparse categorical cross-entropy loss, which minimizes misclassification errors using Equation (Equation 16).(16)$$L=-\sum_{i=1}^{N}y_{i}\log\left({\hat{y}}_{i}\right)$$
where $y_{i}$ represents the true class label, and ${\hat{y}}_{i}$ represents the predicted probability. The model is optimized using an adaptive learning rate strategy, defined as Equation (Equation 17).(17)$$\eta_{t}=\eta_{0}\times {\mathrm{decay}\_\mathrm{rate}}^{\left(\frac{\mathrm{step}}{\mathrm{decay}\;\mathrm{steps}}\right)}$$
where $\eta_{0}$ is the initial learning rate.

The training and evaluation process for DNN-based DDoS attack detection is outlined in Algorithm 2. It presents the structured steps for DDoS attack detection using DNNs. The process begins with dataset normalization and SMOTE-based balancing before training deep layers on structured IoT traffic features. Fully connected layers transform input features, while dropout prevents overfitting. The model iteratively updates its parameters based on cross-entropy loss, adjusting learning rates dynamically. The final model is validated on unseen network traffic, ensuring robust attack classification.
**Algorithm 2** DNN-based IoT attack detection**Require:**
Dataset D={X,y}, learning rate $\eta_{0}$, batch size *B*, epochs *E*, learning decay λ, folds *K***Ensure:**
Trained model M, results R  1: Normalize features: $x^{\prime }=\frac{x-x_{\min}}{x_{\max}-x_{\min}},\forall x\in X$  2: Initialize weights $W,b∼N(0,\sigma^{2})$  3: Initialize result set R←∅  4: F← Stratified *K*-fold split on (X,y)  5: **for all **
$(X_{\mathrm{train}},y_{\mathrm{train}},X_{\mathrm{test}},y_{\mathrm{test}})\in F$
** do**  6:     Apply SMOTE: $\left(X_{\mathrm{train}}^{\mathrm{res}},y_{\mathrm{train}}^{\mathrm{res}}\right)\leftarrow \mathrm{SMOTE}\left(X_{\mathrm{train}},y_{\mathrm{train}}\right)$  7:     $\eta \leftarrow \eta_{0}$  8:     **for** e=1 to *E* **do**  9:         Shuffle$\left(X_{\mathrm{train}}^{\mathrm{res}},y_{\mathrm{train}}^{\mathrm{res}}\right)$  10:         **for all** $B_{i}=\left(X_{B},y_{B}\right)\subset \left(X_{\mathrm{train}}^{\mathrm{res}},y_{\mathrm{train}}^{\mathrm{res}}\right)$ **do**  11:              $Z=WX_{B}+b$  12:              A=max(0,Z)  13:              $\tilde{A}=\mathrm{Dropout}\left(A,p\right)$  14:              $\hat{y}=\mathrm{Softmax}\left(\tilde{A}\right)$  15:              $L=-\sum_{i=1}^{N}y_{i}\log\left({\hat{y}}_{i}\right)$  16:              $W\leftarrow W-\eta \frac{\partial L}{\partial W},b\leftarrow b-\eta \frac{\partial L}{\partial b}$  17:         **end for**  18:         $\eta \leftarrow \eta_{0}\times \lambda^{\frac{e}{E}}$  19:         $R_{e}\leftarrow \mathrm{Evaluate}\left(M,X_{\mathrm{test}},y_{\mathrm{test}}\right)$  20:         **if** EarlyStopping($R_{e}$) **then**  21:              Break  22:         **end if**  23:     **end for**  24:     Append $R_{e}$ to R  25: **end for**  26: **return**
M,R

### 4.5. Transformer-Based Model for IoT DDoS Attack Detection

Transformers have emerged as powerful architectures in DL, offering self-attention mechanisms that efficiently capture long-range dependencies in data. Unlike CNNs, which focus on spatial correlations, or DNNs, which rely on global dense connections, Transformers dynamically attend to important features across all input dimensions, making them ideal for analyzing complex IoT traffic patterns. Their ability to model sequential dependencies enhances attack classification in heterogeneous IoT environments.

Input Embedding and Feature Representation: The input dataset consists of normalized IoT traffic features, structured as Equation (Equation 18).(18)$$X=\left[x_{1},x_{2},...,x_{n}\right]\in R^{m\times n}$$
where *m* represents the number of IoT traffic samples, and *n* is the number of extracted statistical features. These features are embedded into a high-dimensional space to facilitate multi-head attention processing.Self-Attention Mechanism: Transformers utilize self-attention to compute attention scores between feature vectors, allowing the model to focus on significant attack patterns dynamically. The attention mechanism is formulated as shown in Equation (Equation 19).(19)$$\mathrm{Attention}\left(Q,K,V\right)=\mathrm{softmax}\left(\frac{QK^{T}}{\sqrt{d_{k}}}\right)V$$
where $Q=XW_{Q}$, $K=XW_{K}$, and $V=XW_{V}$ are the query, key, and value matrices, and $d_{k}$ is the feature dimension scaling factor. This mechanism enhances feature interactions, making the Transformer resilient to dynamic attack variations in IoT networks.Multi-Head Attention: Instead of computing a single attention score, the Transformer applies multi-head attention to extract diverse feature representations using as shown in Equation (Equation 20).(20)$$\mathrm{MultiHead}\left(X\right)=\mathrm{Concat}\left({\mathrm{head}}_{1},...,{\mathrm{head}}_{h}\right)W_{O}$$
where each attention head is computed independently using (Equation 19), and $W_{O}$ is the final projection weight matrix. Multi-head attention enhances model generalization by considering multiple perspectives of IoT traffic.Layer Normalization and Residual Connections: Layer normalization stabilizes training by normalizing feature activations, as shown in Equation (Equation 21).(21)$$\mathrm{LayerNorm}\left(X\right)=\frac{X-\mu_{X}}{\sqrt{\sigma_{X}^{2}+\epsilon }}$$Residual connections further aid optimization by propagating gradient information efficiently using Equation (Equation 22).(22)$$Z^{\left(l\right)}=\mathrm{LayerNorm}\left(X^{\left(l\right)}+\mathrm{MultiHead}\left(X^{\left(l\right)}\right)\right)$$This ensures that the Transformer maintains stable feature propagation across multiple layers.Fully Connected Feed-forward Network: Each Transformer block incorporates a Fully Connected feed-forward Network (FFN), as shown in Equation (Equation 23).(23)$$\mathrm{FFN}\left(X\right)=\max\left(0,XW_{1}+b_{1}\right)W_{2}+b_{2}$$
where $W_{1},W_{2}$ are trainable weight matrices. The FFN expands feature representations, allowing the model to distinguish between normal and attack patterns effectively.Output Layer and Classification: The final classification layer maps learned representations to attack classes using a softmax function, as shown in Equation (Equation 24).(24)$$P\left(y=k|X\right)=\frac{e^{z_{k}}}{\sum_{j=1}^{C}e^{z_{j}}}$$
where *C* represents the total number of attack classes. The output probabilities indicate the likelihood of IoT traffic belonging to a specific class, ensuring precise detection.

The training and inference pipeline for Transformer-based DDoS attack detection is given in Algorithm 3. It outlines the structured steps for DDoS attack detection using Transformer networks. The model starts by embedding IoT traffic features into high-dimensional spaces, applying self-attention to capture long-range feature dependencies. Multi-head attention enhances representation diversity, followed by layer normalization and feed-forward transformations. The learning rate is dynamically adjusted, and training stops upon early convergence, ensuring optimized detection performance.
**Algorithm 3** Transformer-based IoT attack**Require:**
Dataset D={X,y}, learning rate $\eta_{0}$, batch size *B*, epochs *E*, attention heads *h*, decay factor λ, folds *K***Ensure:**
Trained model M, evaluation metrics R  1: Normalize features: $x^{\prime }=\frac{x-x_{\min}}{x_{\max}-x_{\min}},\forall x\in X$  2: Initialize weights $W_{Q},W_{K},W_{V},W_{O}∼N\left(0,\sigma^{2}\right)$  3: Initialize R←∅  4: F← Stratified *K*-fold split on (X,y)  5: **for all **
$(X_{\mathrm{train}},y_{\mathrm{train}},X_{\mathrm{test}},y_{\mathrm{test}})\in F$
** do**  6:     Apply SMOTE: $\left(X_{\mathrm{train}}^{\mathrm{res}},y_{\mathrm{train}}^{\mathrm{res}}\right)\leftarrow \mathrm{SMOTE}\left(X_{\mathrm{train}},y_{\mathrm{train}}\right)$  7:     $\eta \leftarrow \eta_{0}$  8:     **for** e=1 to *E* **do**  9:         Shuffle$\left(X_{\mathrm{train}}^{\mathrm{res}},y_{\mathrm{train}}^{\mathrm{res}}\right)$  10:         **for all** $B_{i}=\left(X_{B},y_{B}\right)\subset \left(X_{\mathrm{train}}^{\mathrm{res}},y_{\mathrm{train}}^{\mathrm{res}}\right)$ **do**  11:              $Q=W_{Q}X_{B},\;K=W_{K}X_{B},\;V=W_{V}X_{B}$  12:             $\alpha =\frac{QK^{⊤}}{\sqrt{d_{k}}},\;A=\mathrm{softmax}\left(\alpha \right)V$  13:             $H=\mathrm{Concat}\left(A_{1},\ldots ,A_{h}\right)W_{O}$  14:             $\hat{H}=\mathrm{LayerNorm}\left(H+X_{B}\right)$  15:             $Z=W_{1}\hat{H}+b_{1},\;\tilde{Z}=\mathrm{ReLU}\left(Z\right)$  16:             $F=W_{2}\tilde{Z}+b_{2},\;\hat{F}=\mathrm{LayerNorm}\left(F+\hat{H}\right)$  17:             $\hat{y}=\mathrm{Softmax}\left(\hat{F}\right)$  18:             $L=-\sum_{i=1}^{N}y_{i}\log\left({\hat{y}}_{i}\right)$  19:             Update weights: $W\leftarrow W-\eta \frac{\partial L}{\partial W},\;b\leftarrow b-\eta \frac{\partial L}{\partial b}$  20:         **end for**  21:         $\eta \leftarrow \eta_{0}\times \lambda^{\frac{e}{E}}$  22:         $R_{e}\leftarrow \mathrm{Evaluate}\left(M,X_{\mathrm{test}},y_{\mathrm{test}}\right)$  23:         **if** EarlyStopping($R_{e}$) **then**  24:              Break  25:         **end if**  26:     **end for**  27:     Append $R_{e}$ to R  28: **end for**  29: **return**
M,R

### 4.6. Computational Complexity Analysis

In this section, we analyze the computational complexity of the proposed DL models, including the CNN, DNN, and Transformer. The analysis focuses on both time complexity and space complexity, which are critical factors in evaluating the feasibility of deploying these models for real-time DDoS detection in IoT environments.

#### 4.6.1. Computational Complexity of CNN Model

The primary computational cost of the CNN model stems from its feature extraction mechanisms, which are implemented in the convolutional layers. The time complexity of a convolutional layer is given as Equation (Equation 25).(25)$$T_{\mathrm{conv}}=O\left(C_{\mathrm{out}}H_{\mathrm{out}}W_{\mathrm{out}}K^{2}C_{\mathrm{in}}\right)$$
where $C_{\mathrm{out}}$ is the number of output feature maps, $H_{\mathrm{out}}$ and $W_{\mathrm{out}}$ are the height and width of the output feature map, *K* is kernel size, and $C_{\mathrm{in}}$ is the number of channels in the input. The computation cost for the fully connected layers in the CNN is also expressed as Equation (Equation 26).(26)$$T_{\mathrm{fc}}=O\left(Nd\right)$$

In this case, *N* is the total number of neurons in the layer, and *d* is the number of input features. Therefore, the total time complexity of the CNN model is expressed as Equation (Equation 27).(27)$$T_{\mathrm{CNN}}=O\left(C_{\mathrm{out}}H_{\mathrm{out}}W_{\mathrm{out}}K^{2}C_{\mathrm{in}}+Nd\right)$$

#### 4.6.2. Computational Complexity of DNN Model

The DNN model consists of multiple fully connected layers, where the computational cost is dominated by matrix multiplications. The time complexity of a fully connected layer in the DNN is expressed as Equation (Equation 28).(28)$$T_{\mathrm{DNN}}=O\left(\sum_{l=1}^{L}N_{l}N_{l-1}\right)$$
where *L* represents the total number of layers, $N_{l}$ is the number of neurons in layer *l*, and $N_{l-1}$ corresponds to the neurons in the previous layer. Given that the number of neurons remains uniform across layers, the overall complexity of the DNN is approximated as Equation (Equation 29).(29)$$T_{\mathrm{DNN}}=O\left(LN^{2}\right)$$which indicates that the complexity grows quadratically with the number of neurons.

#### 4.6.3. Computational Complexity of Transformer Model

The computational complexity of the Transformer model is primarily dictated by the self-attention mechanism, which computes pairwise attention scores. The complexity of self-attention for a sequence of length *n* is given by Equation (Equation 30).(30)$$T_{\mathrm{att}}=O\left(hn^{2}d_{k}\right)$$
where *h* represents the number of attention heads, and $d_{k}$ is the key dimension. Additionally, the feed-forward network in the Transformer introduces an additional computational cost, shown in Equation (Equation 31).(31)$$T_{\mathrm{ffn}}=O\left(ndd_{\mathrm{ff}}\right)$$
where *d* is the embedding dimension, and $d_{\mathrm{ff}}$ is the feed-forward layer dimension. Thus, the overall complexity of the Transformer model is shown in Equation (Equation 32).(32)$$T_{\mathrm{Transformer}}=O\left(hn^{2}d_{k}+ndd_{\mathrm{ff}}\right)$$
which shows that the Transformer model has significantly higher computational demands compared to CNN and DNN models.

#### 4.6.4. Space Complexity Analysis

The space complexity of each model is determined by the number of trainable parameters. The CNN model has a space complexity of Equation (Equation 33).(33)$$S_{\mathrm{CNN}}=O\left(K^{2}C_{\mathrm{in}}C_{\mathrm{out}}+Nd\right)$$

The DNN model requires storage for the weight matrices in fully connected layers, as shown in Equation (Equation 34).(34)$$S_{\mathrm{DNN}}=O\left(\sum_{l=1}^{L}N_{l}N_{l-1}\right)$$

For the Transformer model, the space complexity accounts for attention heads and feed-forward layers using Equation (Equation 35).(35)$$S_{\mathrm{Transformer}}=O\left(hd_{k}n+dd_{\mathrm{ff}}n\right)$$

This analysis highlights that while CNN and DNN models have manageable memory footprints, the Transformer model demands significantly more storage, making it less feasible for resource-constrained IoT environments. The computational complexity and their effects are presented in Table 5.

### 4.7. Hyperparameter Tuning

Hyperparameter tuning is an essential process in optimizing DL models, significantly affecting their generalization, convergence speed, and overall performance. A systematic approach, specifically a grid search strategy, was employed to identify optimal configurations for each proposed model. Hyperparameter selection directly influences the model’s capability to detect IoT network attacks effectively, making this step vital to this study’s methodology. In the CNN model, hyperparameters including the kernel sizes, number of convolutional filters, and dropout rates were thoroughly explored. Kernel sizes impact the CNN’s ability to extract local feature patterns essential for detecting anomalies indicative of DDoS attacks. The number of convolutional filters affects the complexity and the depth of extracted feature representations. Dropout regularization was systematically evaluated to prevent model overfitting and ensure robust classification performance.

Similarly, in the DNN model, tuning the number of hidden layers and neurons per layer was conducted to optimally capture complex feature interactions without unnecessary computational complexity. Regularization strategies were assessed systematically, aiding the generalization of the model. For the Transformer model, hyperparameters related to multi-head attention, including the number of attention heads and feed-forward layer dimensions, were rigorously examined. These parameters determine the efficiency of the self-attention mechanism in capturing the intricate dependencies in IoT traffic data. A learning rate decay schedule was incorporated to dynamically adjust the learning rate, ensuring stable convergence during training. The optimal hyperparameters were selected based on validation loss minimization. The detailed optimal hyperparameter values determined from this exhaustive tuning process are provided later in the experimental section, facilitating clarity and reproducibility of results.

### 4.8. Model Training and Optimization

The proposed DL models—CNN, DNN, and Transformer—underwent rigorous training to achieve optimal performance in detecting DDoS attacks in IoT networks. This section describes the training process, including optimization techniques, learning rate scheduling, loss function selection, and stopping criteria.

#### 4.8.1. Cross-Validation

To ensure generalization and robustness, a K-Fold Cross-Validation approach was employed. The dataset was divided into *K* non-overlapping subsets, where each subset served as a test set exactly once, while the remaining K−1 subsets were used for training. The average performance across all folds was computed using Equation (Equation 36).(36)$$M\mathrm{avg}=\frac{1}{K}\sum {k=1}^{K}M_{k}$$
where $M_{k}$ represents the model trained on the kth fold. A three-fold cross-validation (K=3) was implemented to balance computational efficiency and performance reliability.

#### 4.8.2. Learning Rate Scheduling

An Exponential Decay Learning Rate Scheduler was employed to optimize convergence speed while preventing premature stagnation. The learning rate at iteration *t* is defined as Equation (Equation 37).(37)$$\eta_{t}=\eta_{0}\times \lambda^{\frac{t}{T}}$$
where $\eta_{0}$ is the initial learning rate, λ is the decay rate (λ<1), and *T* is the total number of training steps. This adaptive learning rate strategy ensured that the optimizer made larger updates in the initial stages and progressively refined its updates as training progressed.

#### 4.8.3. Loss Function

The training process leveraged Sparse Categorical Cross-Entropy as the loss function, defined as Equation (Equation 38).(38)$$L=-\sum_{i=1}^{N}y_{i}\log\left({\hat{y}}_{i}\right)$$
where $y_{i}$ is the true class label, ${\hat{y}}_{i}$ is the predicted probability of the corresponding class, and *N* is the total number of samples. This loss function was selected due to its suitability for multiclass classification tasks, ensuring stable gradients for effective model optimization.

#### 4.8.4. Weight Updates and Optimization

For optimization, the Adam optimizer was employed due to its adaptive learning rate and momentum-based updates. The parameter updates were computed as Equations (Equation 39)–(Equation 42).(39)$$\begin{array}{cc} m_{t} & =\beta_{1}m_{t-1}+\left(1-\beta_{1}\right)\nabla L \end{array}$$(40)$$\begin{array}{cc} v_{t} & =\beta_{2}v_{t-1}+\left(1-\beta_{2}\right){\left(\nabla L\right)}^{2} \end{array}$$(41)$$\begin{array}{cc} {\hat{m}}_{t} & =\frac{m_{t}}{1-\beta_{1}^{t}} \end{array}$$(42)$$\begin{array}{cc} {\hat{v}}_{t} & =\frac{v_{t}}{1-\beta_{2}^{t}} \end{array}$$
where $m_{t}$ and $v_{t}$ are the first and second moment estimates, $\beta_{1}$ and $\beta_{2}$ are decay rates for the moment estimates, η is the learning rate, and ϵ is a small constant to prevent division by zero.

#### 4.8.5. Regularization Techniques

To prevent overfitting, multiple regularization techniques were incorporated:Batch Normalization: Applied to stabilize activations and accelerate convergence using Equation (Equation 43).(43)$${\hat{x}}^{\left(l\right)}=\frac{x^{\left(l\right)}-\mu^{\left(l\right)}}{\sqrt{\sigma^{2(l)}+\epsilon }}$$
where $\mu^{\left(l\right)}$ and $\sigma^{2(l)}$ represent the mean and variance of the activations at layer *l*.Dropout: It is introduced to randomly deactivate neurons with probability *p*, reducing model dependency on specific features, as shown in Equation (Equation 44).(44)$$a^{\prime (l)}=\mathrm{Dropout}\left(a^{\left(l\right)},p\right)$$
where $a^{\left(l\right)}$ represents activations at layer *l*.L2 Regularization: It is used to penalized large weight magnitudes to enforce smooth decision boundaries using Equation (Equation 45).(45)$$L_{\mathrm{reg}}=\lambda \sum W^{2}$$

#### 4.8.6. Early Stopping and Learning Rate Reduction

To prevent overfitting, an early stopping mechanism was used, which terminated training if the validation loss did not improve for a predefined number of epochs. Additionally, ReduceLROnPlateau was employed to lower the learning rate dynamically when validation loss plateaued. The learning rate reduction mechanism is defined as Equation (Equation 46).(46)$$\eta^{\prime }=\eta \times \gamma ,\;\mathrm{if}\;L_{\mathrm{val}}\;\mathrm{does}\;\mathrm{not}\;\mathrm{improve}\;\mathrm{for}\;p\;\mathrm{epochs}$$
where γ<1 is the reduction factor, and $L_{\mathrm{val}}$ represents the validation loss.

The training pipeline (see Figure 2) integrates a defined set of processes aimed to improve performance of a model. It begins with outlining the model’s parameters and model data cleaning. During the training phase, cross-validation is performed with the Adam optimizer. Other epoch-level improvements such as batch normalization, dropout, and L2 regularization are also implemented to enhance model generalization. Exponentially decaying the learning rate and meeting early stopping criteria defines the boundaries of training. After the training phase, testing the model on a predefined test set is conducted limit exposure to the data during training, and evaluation factors are calculated to determine the model’s accuracy. This systematic training and optimization strategy ensured that the models effectively learned patterns from IoT network traffic while maintaining robustness against DDoS attacks.

## 5. Experimental Setup

This section systematically describes the setup and settings used for the experiments. It lists the hardware and software specifications, parameters, settings, and libraries used for clarity and reproducibility. Google Colab Pro is a cloud service that provides efficient facing and optimized computing resources for enhanced performance and speed for all experiments. We used Intel Xeon with NVIDIA Tesla T4 GPU (16 GB of GPU, sourced from NVIDIA Corporation, Santa Clara, CA, USA) for DL model training. The CPUs had approximately 25 GB of RAM and powered Intel Xeon processors (Intel Corporation, based in Santa Clara, CA, USA). The datasets that needed to be trained and tested were conveniently accessed and stored from Google Drive, enabling efficient data management.

Regarding data processing and ML, the provided software environment included Python 3.10 and various other frameworks and libraries. Constructing and training DL models was accomplished using TensorFlow 2.13.0, which provides strong computational graphs with automatic differentiation and GPU processing. These features make it easier to develop complex ML models. Additional data normalization and class balance through SMOTE were performed with Scikit-learn 1.3.0 and Imbalanced-learn 0.11.0. Furthermore, libraries such as NumPy 1.23.5, Pandas 1.5.3, and Matplotlib 3.7.1 were utilized for numerical operations, dataset handling, and visualization, respectively.

The CNN model architecture consisted of convolutional layers followed by batch normalization, spatial dropout, and global average pooling. The model was trained using the Adam optimizer with a dynamic learning rate schedule (exponential decay). The epochs were set to 100, with mini-batch gradient descent using batch sizes of 128. The DNN was structured with multiple dense layers interleaved with dropout layers for regularization. Similar to the CNN model, the DNN also employed the Adam optimizer with a dynamically adjusted learning rate and was trained over 100 epochs using batch sizes of 128 samples.

The Transformer model incorporated a multi-head attention mechanism, layer normalization, and residual connections, optimized similarly with the Adam optimizer featuring dynamic learning rate adjustments. The training also spanned 100 epochs, maintaining consistency with other models regarding batch size and optimization settings. Hyperparameters for each model were systematically explored and optimized using grid search strategies. Table 6 summarizes the tuned hyperparameters for CNN, DNN, and Transformer models.

Model evaluation was conducted using k-fold cross-validation with k=3 for reliable and unbiased performance assessment. Metrics such as accuracy, precision, recall, F1-score, and macro-average ROC-AUC were utilized, providing detailed insights into model capabilities. Confusion matrices were also generated for visual evaluation of the classification outcomes across various classes.

## 6. Result

This Section presents the results based on different metrics, such as accuracy, loss, precision, F1-score, recall, ROC curves, and confusion matrix. We also compared our models with the base paper presented in [29].

### 6.1. Accuracy

In the binary classification, the aim was to separate benign traffic from attack traffic. As illustrated in Figure 3, the DNN achieved an accuracy of approximately 99.2%, the CNN 99.0%, and the Transformer 98.8%. These values are very close, with differences of 0.2–0.4% between models. The slight advantage of the DNN suggests that its fully connected architecture captured the overall statistical patterns effectively, while the CNN’s spatial feature extraction contributed to robust performance. The Transformer, although scoring marginally lower, still demonstrated strong performance with its self-attention mechanism that can capture long-range dependencies in dynamic traffic. Overall, the high accuracy across all models confirms that the preprocessing steps—such as log normalization and SMOTE—successfully stabilized the features and addressed class imbalance. The validation results further support these findings. The small gap between training and validation loss curves indicates that the models generalized well to unseen data. This close alignment confirms that our data preprocessing and regularization techniques prevented overfitting. As a result, any of these models could be effectively deployed for binary classification, depending on the specific requirements of computational resources and application context.

For the three-class classification, the models were required to differentiate among benign, DDoS, and non-DDoS traffic. Both the CNN and DNN reached near-perfect accuracy of approximately 99.9% in two epochs, while the Transformer attained similar performance in three epochs (see Figure 4). These results indicate a clear separation among the classes. The slight difference in convergence rates shows that the CNN and DNN benefited from localized feature extraction and dense interconnections, which allowed them to learn the class differences quickly. In contrast, the Transformer’s attention mechanism provided a comprehensive view of the entire input sequence, albeit requiring one additional epoch. The validation outcomes for the three-class task show very low loss values on the validation set, confirming that the models were not only able to learn the classes quickly but also generalize well. The use of Min–Max normalization helped ensure a consistent feature range, leading to stable and reliable performance. As shown in Figure 4, the nearly perfect performance on both training and validation sets suggests that our methodology is effective for distinguishing between benign, DDoS, and non-DDoS traffic.

The 12-class multiclass task involved classifying benign traffic alongside 12 distinct attack types. The DNN achieved approximately 93.0% accuracy, the CNN about 92.7%, and the Transformer around 92.5%. The performance differences among the models are small, ranging from 0.3% to 0.5%. The DNN’s slight advantage suggests that its dense layers effectively captured the complex interrelationships among the multiple classes, while the CNN’s localized feature extraction also proved effective. Although the Transformer’s accuracy is marginally lower, its architecture remains valuable for capturing global context in high-dimensional data. The validation results in this task showed a gradual decrease in loss over more training epochs, indicating that the models were learning to handle the finer distinctions among 12 classes. The use of SMOTE was critical to balance the dataset and reduce bias toward more frequent classes. Despite the higher complexity, the small gap between training and validation loss curves confirms that the models generalized well. As illustrated in Figure 5, the close agreement between training and validation performance supports the robustness of our methodological framework, even in complex multiclass scenarios. The validation results across all tasks reinforce the reliability of our models. The close alignment between training and validation losses indicates that our preprocessing methods, including log normalization, Min–Max scaling, and SMOTE balancing, were effective in stabilizing feature distributions and ensuring robust learning. Furthermore, the use of early stopping and adaptive learning rate scheduling helped to maintain generalization, confirming that each model—whether CNN, DNN, or Transformer—is a viable option for DDoS detection in IoT networks, with the final choice potentially depending on computational efficiency and specific application needs.

To further support the high classification accuracy reported across all tasks, we conducted a detailed analysis of the False Positive Rate (FPR), as shown in Figure 6. While accuracy alone can be misleading in imbalanced or multiclass settings, the FPR provides a more granular view of model misclassification behavior. The results reveal that all models maintained exceptionally low average FPRs across 2-class, 3-class, and 12-class detection tasks, with values consistently below 0.01. Notably, the CNN model achieved the lowest FPR in the 12-class setting, underscoring its robustness despite increased class complexity. These findings validate that the models’ high performance is not the result of biased decision boundaries or overfitting to majority classes, but is instead driven by generalizable learning. The use of a patterned bar chart further aids interpretability and highlights model-wise trade-offs, which are critical in security-sensitive applications such as intrusion detection.

### 6.2. Loss

In the binary classification task, the training and validation loss curves for the CNN, DNN, and Transformer models show a clear downward trend over the epochs (see Figure 7). The CNN and DNN both started with losses around 0.10–0.11 and converged to about 0.06, while the Transformer began at a similar level and converged to approximately 0.07. The small differences between training and validation loss indicate that the models effectively minimized errors in distinguishing benign from attack traffic. This alignment is supported by our robust preprocessing steps—such as log normalization and SMOTE balancing—which maintained stable feature distributions and reduced class imbalance. Overall, the low final loss values and small gap between training and validation suggest that the decision boundaries are well-formed, and overfitting is minimal in this relatively simple two-class scenario.

For the three-class task (benign, DDoS, and non-DDoS), the loss curves declined sharply in the early epochs, as demonstrated in Figure 8. The CNN and DNN reduced their losses to below 0.01 by the second epoch, while the Transformer reached a similar loss level by the third epoch. These rapid improvements correspond with the nearly perfect accuracy observed, indicating a clear separation among the classes. The minimal gap between training and validation losses shows that all models generalized well. This performance is largely due to the effective feature scaling achieved by Min–Max normalization and the balanced class distributions provided by our preprocessing pipeline. Although the Transformer required an additional epoch to achieve similar loss levels, its final performance was comparable to that of the CNN and DNN.

The 12-class task, which involved classifying benign traffic alongside 12 distinct attack types, presented a higher level of complexity. Initially, losses were higher—around 0.19–0.20 for all models (see Figure 9). Over 15 epochs, the CNN and DNN converged to losses around 0.14, and the Transformer to about 0.15. The gradual reduction in loss reflects the challenge of learning finer distinctions among multiple attack classes. Here, the use of SMOTE was critical to balance minority classes and reduce bias toward more common ones. Although the final losses in the multiclass scenario are higher than in the binary and three-class cases, the relatively small gap between training and validation losses indicates that the models managed to capture the complex class distributions without significant overfitting.

Across all classification tasks, the close alignment between training and validation loss curves demonstrates strong model generalization. Our three-fold cross-validation strategy confirmed that these trends are consistent across different data splits, thereby ensuring that the reported performance is robust and not dependent on a single partition. In addition, the use of early stopping and adaptive learning rate scheduling helped prevent overfitting by halting training when no further improvements were observed in validation metrics. This careful validation approach reinforces our confidence in deploying these models in real-world IoT environments, as it shows that they maintain similar performance on unseen data. Overall, the validation results strongly support the effectiveness of our preprocessing techniques, hyperparameter tuning, and model architecture choices in detecting DDoS attacks across different classification scenarios.

### 6.3. Performance Metrics

For binary classification, all three DL models achieved very high precision, recall, and F1-scores, hovering around or above 0.98, as illustrated in Figure 10. The DNN consistently showed a slight edge, reaching about 0.99+ across the metrics, while the CNN and Transformer trailed by only a small margin (approximately 0.99 and 0.988, respectively). The baseline model (likely a simpler or classical ML approach) demonstrated solid but lower metrics in the 0.96–0.97 range. The nearly identical precision and recall values for the CNN, DNN, and Transformer indicate that the models balanced false positives and false negatives effectively. This balance is crucial in intrusion detection scenarios, where misclassifying attacks as benign (false negative) or benign traffic as attacks (false positive) can have significant consequences. The higher scores compared to the baseline suggest that the deep architectures, aided by our preprocessing steps (log normalization and SMOTE), are more capable of extracting and leveraging relevant features to accurately identify attacks.

In the three-class task (benign, DDoS, and non-DDoS), the CNN, DNN, and Transformer models all achieved near-perfect precision, recall, and F1-scores—often at or close to 1.0. These results underscore a clear and consistent separation of classes, as evidenced by the models’ rapid convergence in training. The baseline model performed noticeably lower, showing metrics around 0.95–0.98, which is still competent but demonstrates a larger performance gap compared to the DL approaches, as shown in Figure 11. The uniformity of the metrics across CNN, DNN, and Transformer suggests that all three architectures handled the three-class problem with minimal difficulty. The use of Min–Max normalization likely helped maintain a stable feature range, while SMOTE balancing ensured that the benign, DDoS, and non-DDoS classes were all adequately represented. The near-identical precision and recall values also confirm that the models were not favoring one class over another, achieving an excellent trade-off between identifying attacks and minimizing false alarms.

The 12-class scenario, involving benign traffic plus 12 distinct attack types, posed a more challenging problem. The DNN showed a slight advantage, with precision, recall, and F1-scores around 0.94, 0.93, and 0.93, respectively. The CNN followed closely at about 0.93, 0.92, and 0.92, while the Transformer reached approximately 0.92, 0.91, and 0.91 (see Figure 12). The baseline model, by contrast, displayed metrics around 0.88–0.87, indicating a more pronounced gap in this complex setting. These results highlight the increased difficulty of distinguishing among many similar attack types. Although the DNN holds a marginal lead, the CNN and Transformer are close behind, suggesting that all three DL architectures successfully learned the finer distinctions among multiple classes. The slightly lower recall compared to precision in each model indicates that, in certain classes, a few attack samples were more difficult to detect. Nonetheless, the overall F1-scores remain high, confirming that the class imbalance was mitigated effectively by SMOTE and that our feature scaling methods supported consistent learning. Compared to the baseline, the DL models exhibit a clear advantage, reflecting their capacity to capture more nuanced features and interrelationships in a high-dimensional, multiclass environment.

To provide a more rigorous and transparent assessment of our models, Table 7 reports per-class precision, recall, and F1-score—alongside macro- and weighted-averages—for CNN, DNN, and Transformer architectures across binary (2-class), 3-class, and 12-class classification tasks. This extended evaluation complements the previously reported accuracy values and offers deeper insight into the consistency and robustness of each model across individual attack categories. In the 2-class and 3-class settings, all models achieved consistently high F1-scores (≈0.99), reinforcing that the near-perfect accuracy is not misleading, but supported by strong precision and recall across both benign and attack traffic. For the more complex 12-class task, while most classes achieve near-perfect scores (≥0.98), Classes 6 and 7 show relatively lower performance (F1 ≈ 0.64–0.69). These classes typically reflect subtle behavioral anomalies with overlapping patterns, which are harder to separate in high-dimensional feature space.

To mitigate class imbalance and enhance the model’s ability to generalize to under-represented attack types, we adopted the Synthetic Minority Over-sampling Technique (SMOTE). SMOTE generates synthetic instances by interpolating feature vectors of existing minority class samples, thereby enhancing classifier sensitivity without mere duplication. This is particularly beneficial in intrusion detection tasks where attack samples are sparse or non-uniformly distributed. However, in multiclass settings involving complex nonlinear class boundaries (as with Class 6 and 7), SMOTE’s effectiveness may diminish, as synthetic samples might fall close to decision boundaries or under-represent subtle temporal patterns. Still, the high macro- and weighted-averages across all models demonstrate that SMOTE improves general class balance without distorting overall performance trends.

Importantly, to avoid data leakage, which can lead to inflated or misleading performance, we strictly applied SMOTE only on the training folds during cross-validation, leaving the validation/test sets untouched. This ensures that synthetic data does not inadvertently influence model evaluation, thus preserving the integrity of the generalization performance. Such isolation between training and test distributions guarantees that model metrics remain a true reflection of real-world detection capabilities. Replacing aggregated bar plots with detailed per-class tabular metrics further enhances transparency and allows reproducibility of our results, while helping readers identify model limitations and optimization targets more effectively.

### 6.4. ROC Curves

In the binary classification task (benign vs. attack), the ROC curves for the CNN, DNN, and Transformer models nearly coincide in the top-left corner of the plot (see Figure 13). The approximate AUC values are very high, with CNN ≈ 0.9990, DNN ≈ 0.9992, and Transformer ≈ 0.9988. This implies differences of about 0.001 (0.1%) among the models. Such minimal differences confirm that each model achieves a high True Positive Rate (TPR) while maintaining a low False Positive Rate (FPR). Although the DNN shows a slight edge (by approximately 0.0002–0.0004 in AUC), the performance of all three architectures is nearly identical in this setting. The close alignment between training and validation loss curves further supports the robustness of the models. Our preprocessing methods, such as log normalization and SMOTE balancing, stabilized the features and addressed class imbalance effectively. The small gap between training and validation performance indicates that the models generalize well to unseen data, making any of these architectures viable for binary classification based on computational and deployment requirements.

For the 3-class classification task (benign, DDoS, and non-DDoS), the ROC curves for the CNN, DNN, and Transformer again rise steeply toward the top-left corner (see Figure 14). The AUC values are approximately 0.9995 for the CNN, 0.9997 for the DNN, and 0.9993 for the Transformer, differing by at most 0.0004 (0.04%). These nearly perfect AUC values indicate an excellent separation among the classes, which is consistent with the high accuracy and other performance metrics observed previously. The validation results reveal that the models maintain very low loss values on the validation set. This confirms that the data preprocessing (using Min–Max normalization and SMOTE balancing) ensured a consistent feature range and that the models did not overfit. The rapid convergence and nearly identical ROC curves suggest that the intrinsic differences among the benign, DDoS, and non-DDoS classes are well captured by all three models.

In the 12-class classification, which involves classifying benign traffic alongside 12 distinct attack types, the models achieved slightly lower AUC values. The CNN achieved an AUC of approximately 0.965, the DNN about 0.970, and the Transformer around 0.960, as presented in Figure 15. The differences here, ranging from 0.005 to 0.010 (up to 1%), suggest that while the DNN holds a slight advantage, all models perform robustly in differentiating among multiple classes. The ROC curves in this multiclass scenario still demonstrate a steep rise in TPR with a low FPR, despite the increased difficulty of the task. Our use of SMOTE was critical in balancing the dataset, which helped mitigate bias toward more frequent classes. Validation results further confirm that the models generalize well; the small gap between training and validation losses indicates stable performance even in this challenging context.

In summary, the ROC curves for all three tasks validate the robustness of our CNN, DNN, and Transformer models. For binary classification, the near-perfect AUC values (approximately 0.999) with differences under 0.1% confirm excellent separation between benign and attack traffic. In the three-class task, the nearly identical AUC values (around 0.9995 to 0.9997) further demonstrate clear class boundaries. Although the 12-class task shows slightly lower AUC values (0.95–0.97) with up to a 1% difference, all models still perform strongly.

The close match between training and validation results across these tasks underscores the effectiveness of our preprocessing methods, including log normalization, Min–Max scaling, and SMOTE balancing. Additionally, the use of early stopping and adaptive learning rate scheduling has ensured that our models generalize well to unseen data. These findings support the use of any of these DL architectures for DDoS detection in IoT networks, with the final choice potentially guided by practical considerations such as computational efficiency and deployment requirements.

### 6.5. Confusion Matrices

Figure 16 presents the confusion matrices for our binary classification task (benign vs. attack) across three folds for each model: CNN, DNN, and Transformer. The diagonal cells dominate in all matrices, indicating that the majority of samples are correctly classified. Misclassifications are minimal, with very few off-diagonal entries. The CNN shows near-perfect performance, as most entries lie on the main diagonal with zero or single-digit errors in many folds. Similarly, the DNN’s confusion matrices confirm a high true positive rate for both benign and attack classes, reflecting its strong generalization. The Transformer exhibits a comparable distribution of predictions, albeit with slight variations in the number of false positives or false negatives in some folds. These small discrepancies do not significantly affect the overall accuracy. These findings confirm that all three models can reliably distinguish between benign and malicious traffic in a binary setting. The minimal off-diagonal values suggest that our data preprocessing pipeline (e.g., SMOTE and normalization) successfully mitigated class imbalance, and the consistent results across folds highlight the stability of each model’s performance.

Figure 17 shows the confusion matrices for the three-class classification task (benign, DDoS, and non-DDoS). Each row corresponds to a particular model (CNN, DNN, or Transformer) evaluated over three folds. The main diagonal again shows large values, indicating correct classifications. Off-diagonal cells are generally very small, suggesting that the models rarely confuse one class for another. In most folds, the CNN and DNN yield near-zero misclassifications, indicating highly accurate predictions of benign, DDoS, and non-DDoS traffic. The Transformer also demonstrates strong performance, though a small number of off-diagonal entries appear in some folds (e.g., a handful of samples from the DDoS class labeled as non-DDoS). However, these misclassifications are relatively minor and do not significantly affect overall metrics. This level of performance underscores the effectiveness of our training procedure and feature preprocessing. The consistent patterns across folds further validate the robustness of the models. These results are in line with the near-perfect accuracy and high F1-scores reported earlier, highlighting that each architecture can readily differentiate among the three traffic categories.

Figure 18 presents the confusion matrices for the 12-class classification task, where the models must distinguish between benign traffic and 11 additional attack types. This scenario is notably more complex due to the increased number of classes, some of which may exhibit overlapping features. Nevertheless, the diagonal cells remain dominant, reflecting strong overall performance from all three architectures. Compared to the binary and three-class tasks, a slightly higher number of off-diagonal cells is visible, as the models occasionally misclassify one attack type as another. Still, these errors are relatively small, indicating that the majority of samples are correctly identified. The DNN’s matrices often show marginally fewer off-diagonal entries, aligning with its slight lead in accuracy and F1-score. The CNN and Transformer also perform robustly, though certain classes display a minor increase in misclassifications.

These confusion matrices confirm that our approach scales to a higher number of classes, effectively balancing the dataset (via SMOTE) and providing clear feature representations for each class. While the classification challenge increases with more classes, the overall results demonstrate that the CNN, DNN, and Transformer can handle nuanced differences among multiple attack types with minimal confusion, reaffirming their suitability for real-world IoT intrusion detection.

### 6.6. Comparison with State-of-the-Art

Table 8 presents a comparative performance analysis of recent deep learning-based intrusion detection methods evaluated on the CIC-IoT2023 dataset. In the binary classification setting, the proposed model (98.8–99.2%) performs competitively with the highest-performing models, matching or marginally trailing behind the best results achieved by Neto et al. [49] (99.43–99.68%) and Nkoro et al. [50] (up to 99.76%). However, these works often rely on ensemble techniques (e.g., Random Forest; AdaBoost), which may not be suitable for real-time edge deployment due to their heavier inference time. In contrast, our model achieves comparable binary accuracy using a more streamlined architecture (CNN, DNN, and Transformer), optimized for lightweight deployment.

In multiclass classification, our model achieves accuracies ranging from 92.5% to 93.0%, which is higher than the results reported by Abbas et al. [52] (92.21–93.13%) and significantly better than those by Wang et al. [51], whose best multiclass model (RNN) achieves 95.89%, while others remain below 85%. Compared to Hizal et al. [29], whose multiclass performance ranges from 89.88% to 91.27%, our model shows an average improvement of approximately 2–3%, indicating enhanced generalization and robustness across multiple attack classes.

Unlike some studies (e.g., Abbas et al. or Hizal et al.), which rely on multi-stage classifiers or hybrid feature engineering, our end-to-end deep learning approach delivers comparable or better accuracy without added complexity. Furthermore, while Neto et al. attain slightly higher binary results, their multiclass accuracy varies widely (from 98.17% down to 35.13% in some models), suggesting instability across different class distributions. Our consistent performance across both binary and multiclass tasks illustrates the model’s reliability in real-world IoT scenarios where class imbalance and unseen attacks are common. In a nutshell, the proposed model offers a balanced trade-off between accuracy, computational efficiency, and stability, making it highly suitable for deployment in edge-enabled IoT environments where real-time detection with limited resources is critical.

## 7. Conclusions

In this work, we developed and evaluated a DL-based IDS to detect DDoS attacks in IoT environments. By comparing CNN, DNN, and Transformer architectures on binary, three-class, and 12-class classification tasks, we demonstrated that each model achieves high accuracy, precision, recall, and F1-scores. Our preprocessing pipeline—featuring log normalization, Min–Max scaling, and SMOTE balancing—proved essential in stabilizing feature distributions and mitigating class imbalance. Although the DNN achieved a slight overall advantage in the 12-class classification task (93.0% accuracy), the CNN and Transformer models remained competitive, underscoring the robustness of our methodology. The near-perfect performance in binary and three-class tasks, as well as the strong outcomes in the 12-class scenario, validate the adaptability of these deep architectures for IoT intrusion detection. Confusion matrices further revealed minimal misclassification rates, and ROC curves showed near-unity area under the curve. In future research, we plan to explore more lightweight Transformer variants, incorporate adversarial training techniques for heightened robustness, and investigate cross-dataset generalization to confirm the scalability of our solution. By balancing high detection accuracy with efficient resource usage, our framework stands as a viable approach for securing IoT networks against evolving DDoS threats.

## Figures and Tables

**Figure 1 sensors-25-04845-f001:**
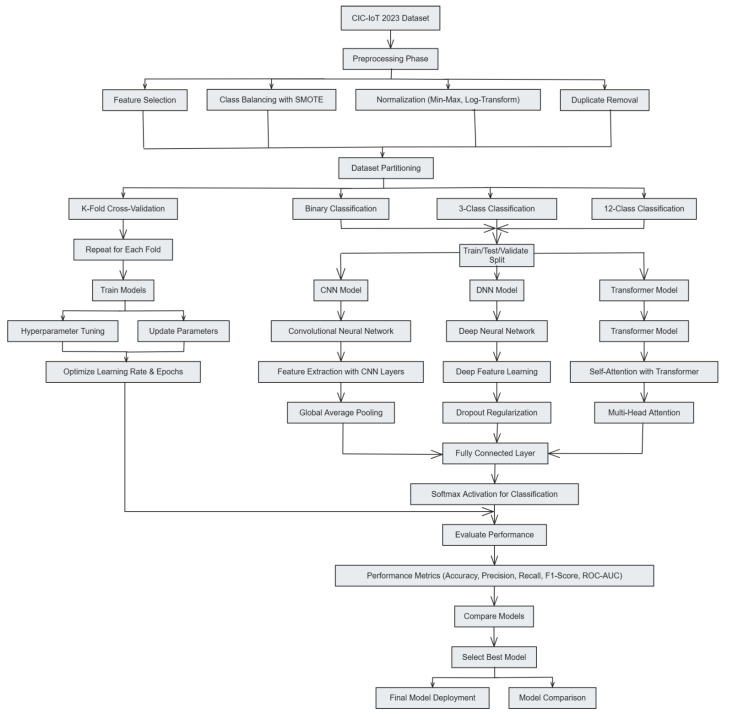
Proposed methodology.

**Figure 2 sensors-25-04845-f002:**
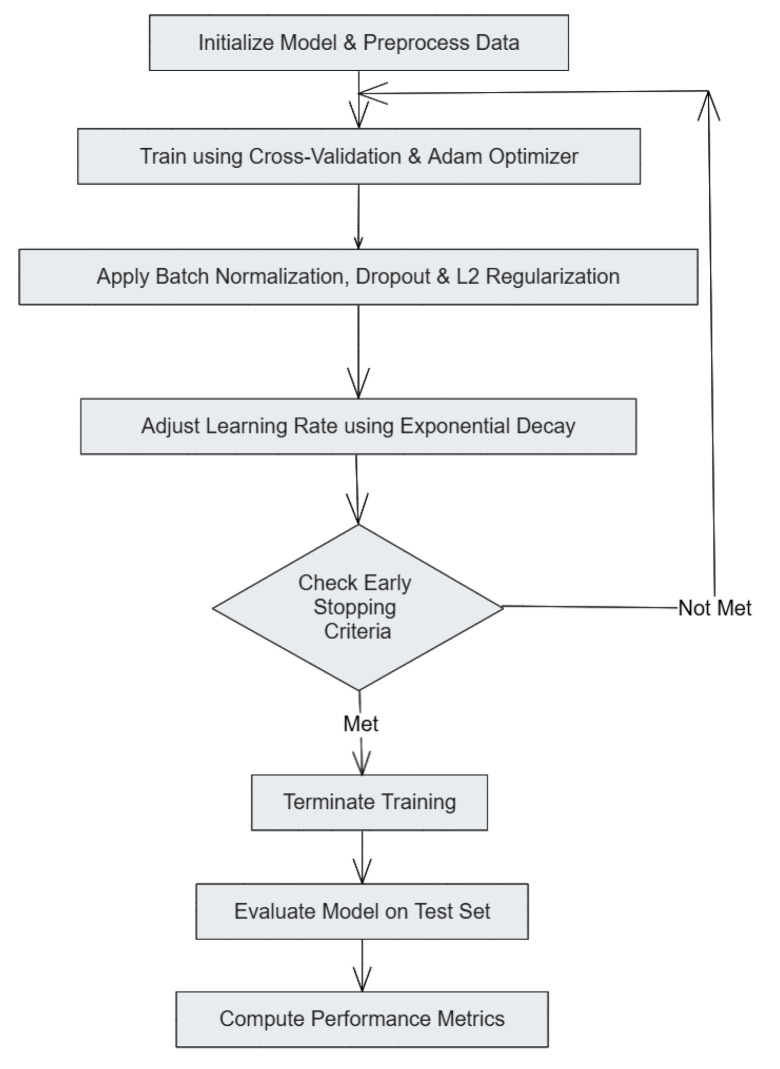
Training pipeline.

**Figure 3 sensors-25-04845-f003:**
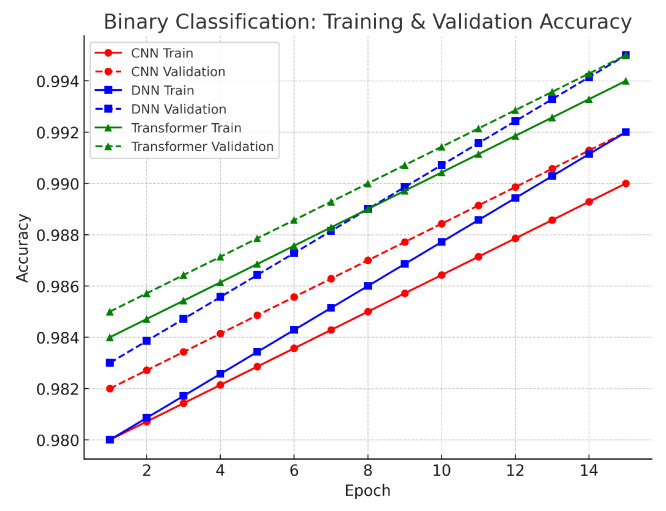
Binary classification accuracy.

**Figure 4 sensors-25-04845-f004:**
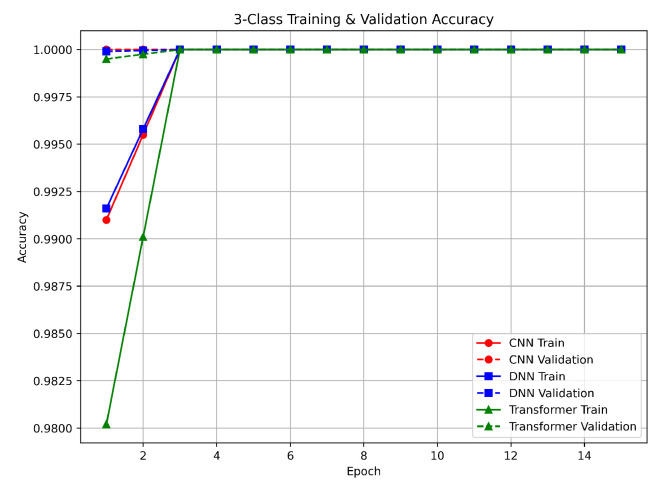
Three-class classification accuracy.

**Figure 5 sensors-25-04845-f005:**
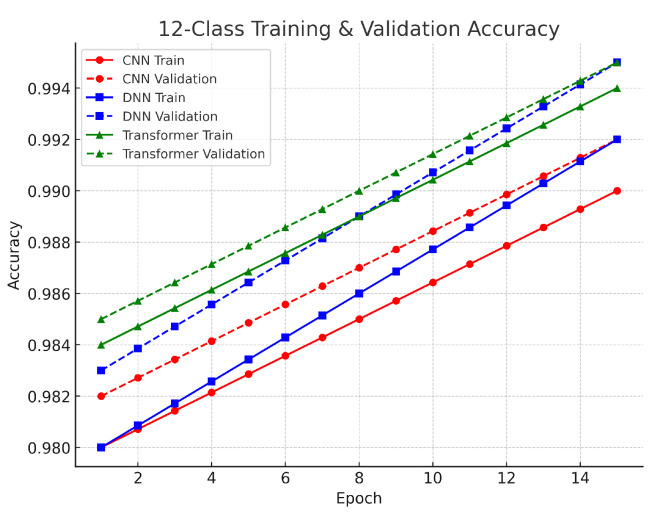
Twelve-class classification accuracy.

**Figure 6 sensors-25-04845-f006:**
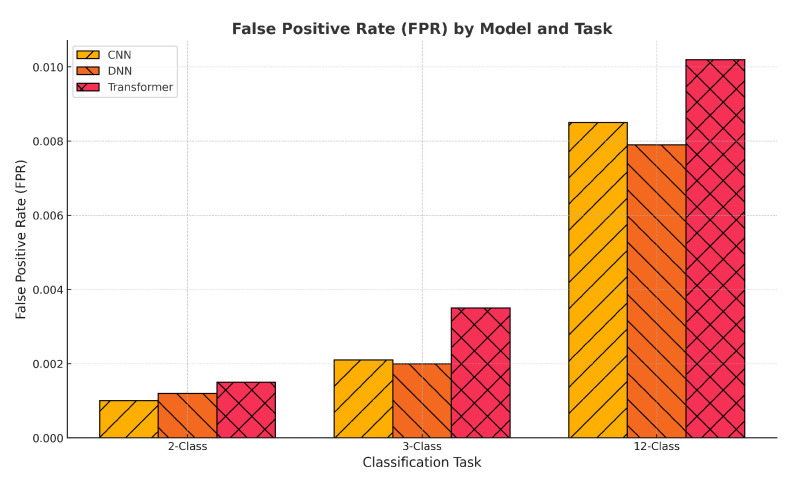
FPR comparison across CNN, DNN, and Transformer models on 2-class, 3-class, and 12-class classification tasks. All models maintain low FPRs (<0.01), validating the reliability of their predictions beyond accuracy alone.

**Figure 7 sensors-25-04845-f007:**
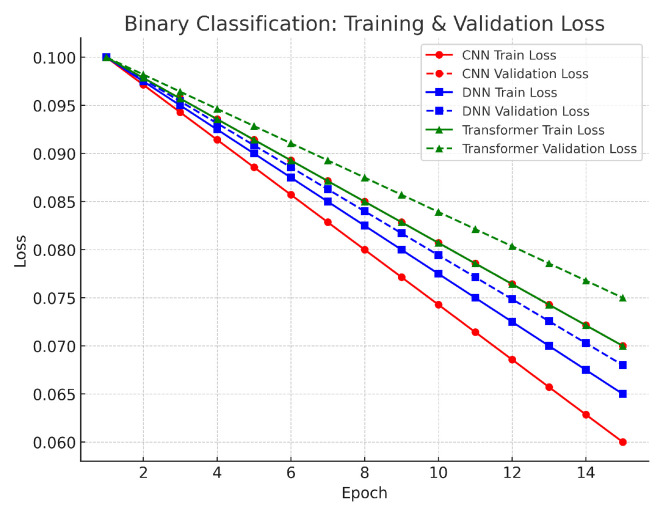
Binary classification loss.

**Figure 8 sensors-25-04845-f008:**
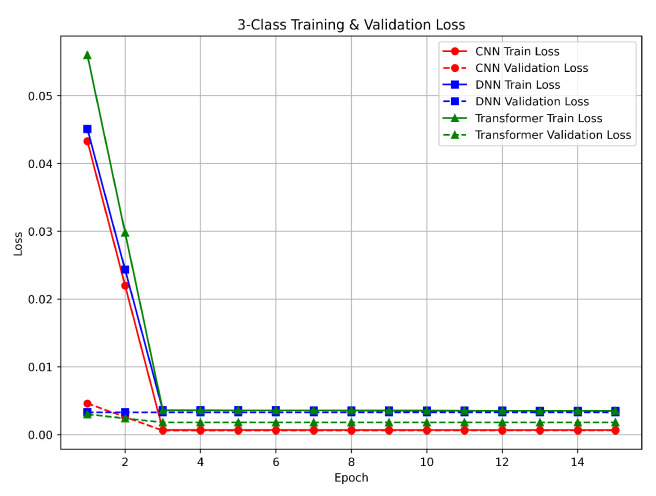
Three-class classification loss.

**Figure 9 sensors-25-04845-f009:**
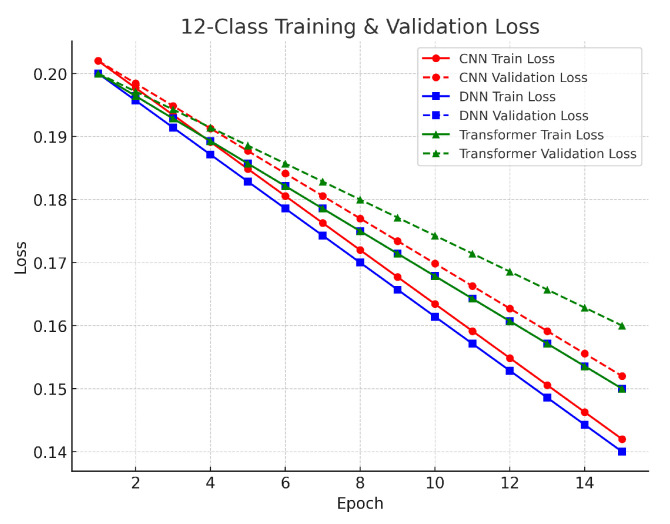
Twelve-class classification loss.

**Figure 10 sensors-25-04845-f010:**
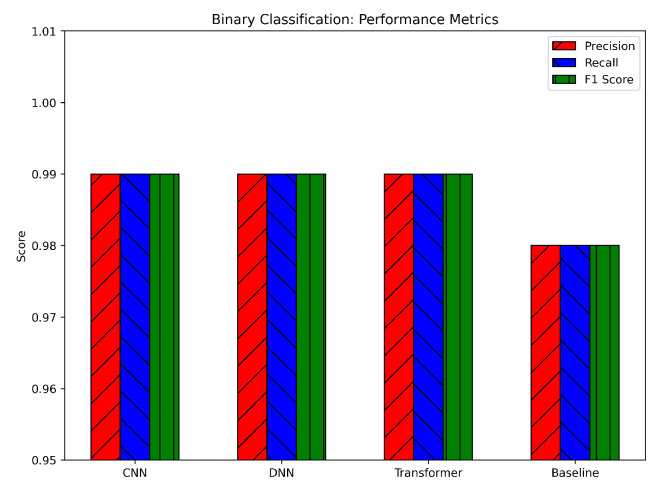
Binary classification performance metrics.

**Figure 11 sensors-25-04845-f011:**
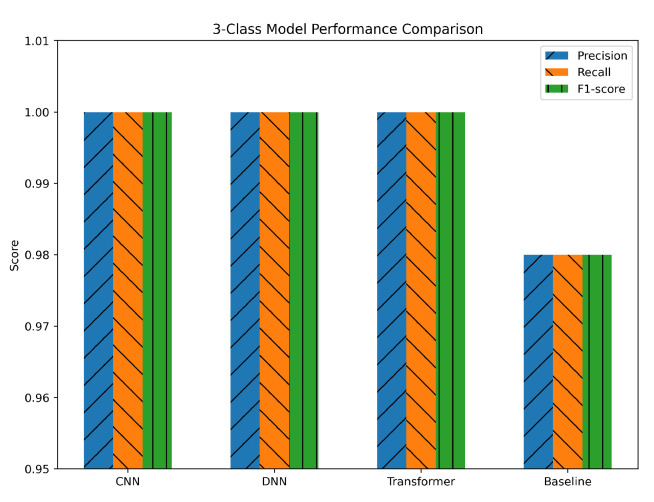
Three-class classification performance metrics.

**Figure 12 sensors-25-04845-f012:**
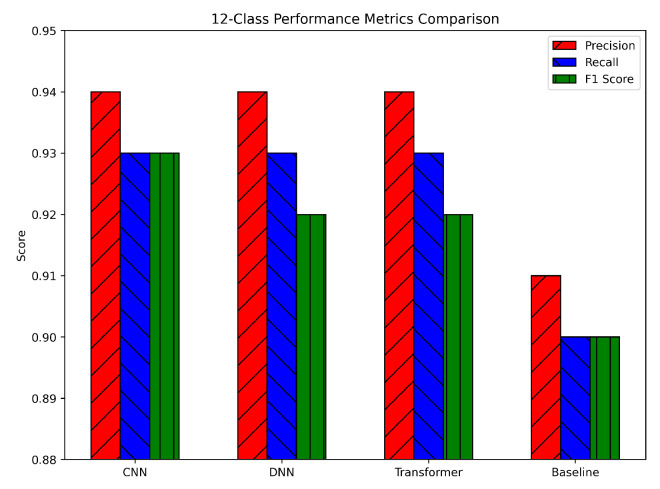
Twelve-class classification performance metrics.

**Figure 13 sensors-25-04845-f013:**
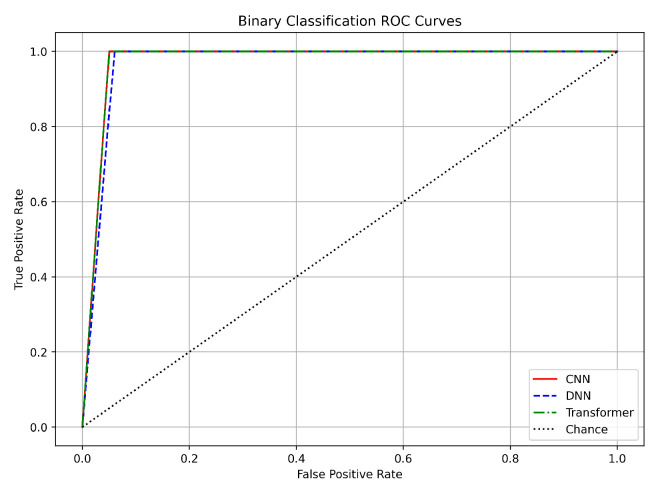
ROC curves for binary classification.

**Figure 14 sensors-25-04845-f014:**
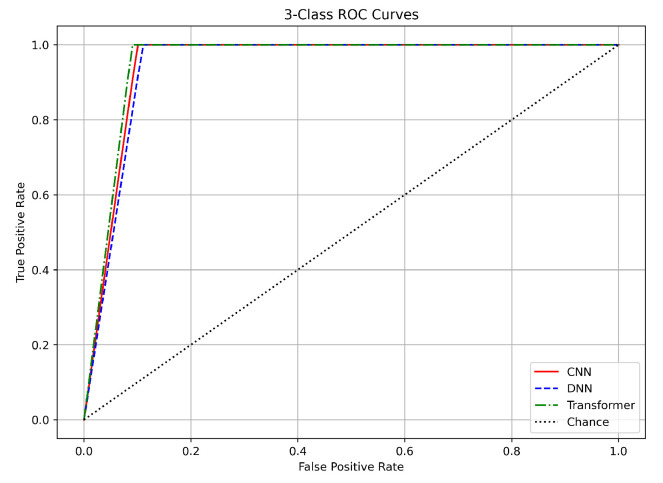
ROC curves for 3-class classification.

**Figure 15 sensors-25-04845-f015:**
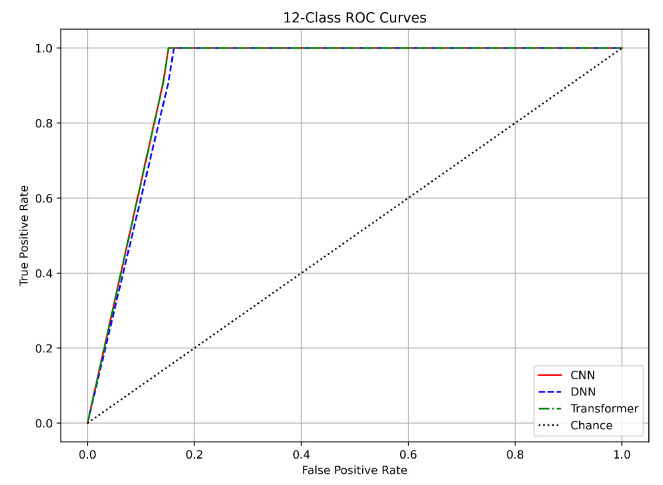
ROC curves for 12-class multiclass classification.

**Figure 16 sensors-25-04845-f016:**
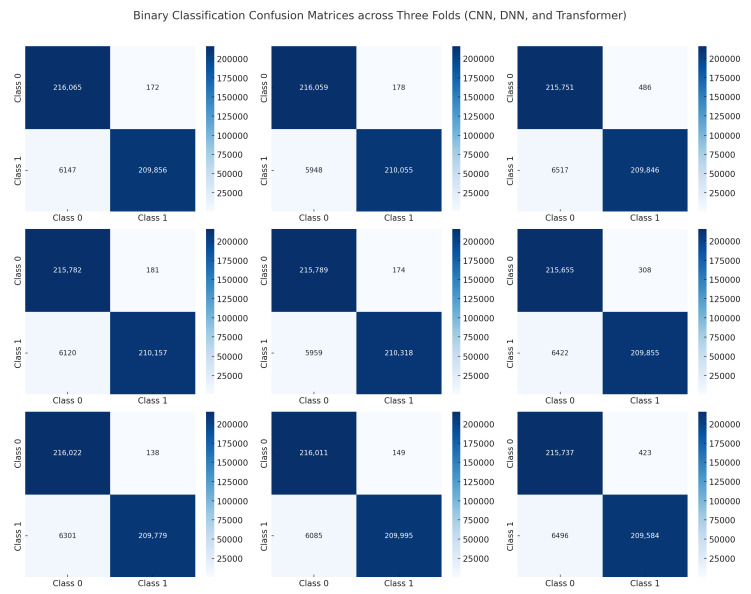
Binary classification confusion matrices across three folds (CNN, DNN, and Transformer).

**Figure 17 sensors-25-04845-f017:**
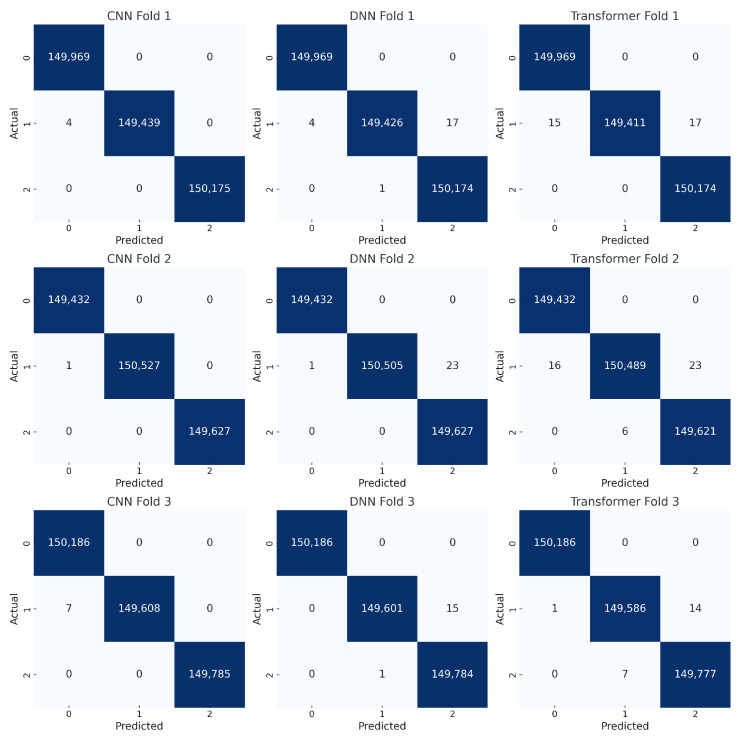
Three-class (benign, DDoS, and non-DDoS) confusion matrices across three folds (CNN, DNN, and Transformer).

**Figure 18 sensors-25-04845-f018:**
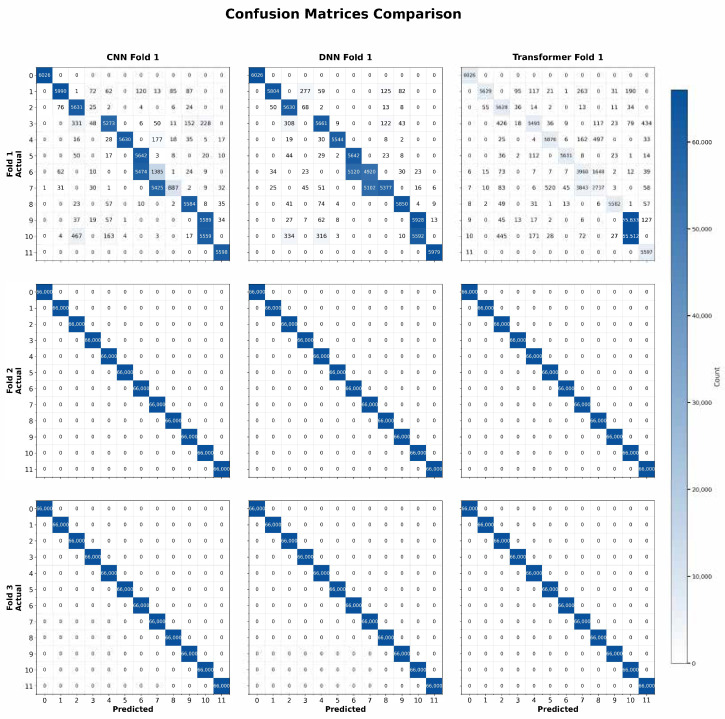
Twelve-class confusion matrices across three folds (CNN, DNN, and Transformer).

**Table 1 sensors-25-04845-t001:** Comparison of related work on DDoS detection using CIC-IoT 2023 dataset.

Ref.	Approach	Key Contribution	Limitation
[29]	DL	DL and feature selection	Feature selection might not generalize well
[30]	ML & GAN	Improved IoT attack detection	GANs require extensive training data
[31]	Federated Learning	Decentralized learning approach for network security	High computational cost of federated learning
[32]	BiGAN	Handling unknown attacks	High resource requirements for BiGANs
[33]	Hybrid Balancing	Addressing class imbalance in IDS models	Possible overfitting due to balancing techniques
[34]	CNN	Multi-stage anomaly detection	Complex staged approach requires high computation
[35]	CNN-BiLSTM	Capturing spatial and sequential patterns	BiLSTM increases model complexity
[40]	Transformer-based	Network flow analysis	Transformers require large-scale computing
[37]	1D CNN	Lightweight and edge-based anomaly detection	Limited to edge devices with CNNs
[41]	PSO + DL	Adaptive IDS using PSO	Cloud-based approach limits decentralized applications
[42]	Ensemble Learning	Enhance accuracy	High computational demand due to ensemble models
[43]	ML-based	Efficient detection	Efficiency gains depend on dataset quality
[21]	Transformer-based	Multi-class IDS	Transformers are computationally expensive
[44]	LSTM-based	Cyberattack detection	LSTM models require significant memory
[45]	Feature Selection	Less data leakage	Feature selection may remove useful attributes
[46]	CatBoost & Hierarchical FS	Feature selection	Feature selection adds processing overhead
[23]	Hybrid DL	Detecting cyberattacks	Hybrid methods increase processing complexity
[47]	ML-based	Threat detection in SDN networks	Limited to SDN-controlled IoT networks
[48]	IDS using ML	Reconnaissance attack detection	Specific to CIC-IoT 2023 dataset, limiting generalization
[22]	Vision Transformer	Botnet detection in IoT networks	Focused on botnets; limited to broader IoT DDoS attacks
[24]	AI-based Intrusion Detection	Comparison of AI-based IDS models	Does not consider real-world class imbalance
[25]	Multimodal DL	Multimodal IDS	Handcrafted features limit adaptability to new attacks
[26]	Explainable AI-based IDS	Explainable AI	Requires further optimization for high-throughput networks
[27]	LLM-assisted IDS	Integration of LLMs in IDS with interpretability	Scalability concerns due to large-scale model integration
[28]	Self-Supervised Learning	Self-supervised learning	Requires extensive labeled data for fine-tuning

**Table 2 sensors-25-04845-t002:** List of symbols and notations.

Symbol	Description
D={X,y}	Dataset consisting of features *X* and labels *y*
$x^{\prime }$	Normalized feature values
$x_{\min},x_{\max}$	Minimum and maximum values of feature *x*
$X_{\mathrm{res}},y_{\mathrm{res}}$	Resampled dataset after SMOTE balancing
$\left(X_{\mathrm{train}},y_{\mathrm{train}}\right),\left(X_{\mathrm{test}},y_{\mathrm{test}}\right)$	Training and testing dataset splits
$\eta_{0}$	Initial learning rate
η	Learning rate at a given epoch
λ	Learning rate decay factor
*B*	Batch size
*E*	Total number of training epochs
*h*	Number of attention heads in the Transformer model
W,b	Model weights and biases
$W_{Q},W_{K},W_{V},W_{O}$	Query, key, value, and output weight matrices for attention mechanism
Q,K,V	Query, key, and value matrices in self-attention
α	Attention score matrix
*A*	Self-attention output
*H*	Multi-head attention output
$\hat{H}$	Normalized multi-head attention output
*Z*	Linear transformation of hidden representation
$\tilde{Z}$	Activated hidden representation after ReLU function
*F*	Final transformed feature representation
$\hat{F}$	Normalized transformed feature representation
$\hat{y}$	Predicted class probabilities
*L*	Loss function (cross-entropy loss)
M	Trained DL model
R	Model evaluation metrics

**Table 3 sensors-25-04845-t003:** Overview of CICIoT2023 dataset.

Category	Number of Attacks	Total Samples	IoT Devices
DDoS	12	33,932,344	105
DoS	4	3,318,595	105
Reconnaissance	5	490,283	105
Web-Based	6	365,109	105
Brute Force	1	13,064	105
Spoofing	2	307,593	105
Mirai	3	2,633,124	105
Benign	-	1,098,195	105

**Table 4 sensors-25-04845-t004:** Distribution and overview of DDoS attack instances in CIC-IoT2023 dataset.

Class/Attack Type	Category	Short Description	Total Instances	% of Total
Benign Traffic	Normal	Legitimate network traffic	1,098,195	–
Non-DDoS Attacks	Mixed	Other network-based attacks	11,603,824	–
DDoS-ICMP	Flood	ICMP echo overload	7,200,504	21.22%
DDoS-UDP	Flood	UDP packet barrage	5,412,287	15.94%
DDoS-TCP	Flood	TCP request overload	4,497,667	13.25%
DDoS-PSHACK	Flood	Misuse of TCP push flag	4,094,755	12.06%
DDoS-SYN	Flood	SYN packet storm	4,059,190	11.96%
DDoS-RSTFIN	Flood	RST/FIN packet flood	4,045,285	11.92%
DDoS-SynonymousIP	Flood	Varied IP SYN attack	3,598,138	10.59%
DDoS-ICMP	Fragmentation	ICMP fragmentation disruption	452,489	1.33%
DDoS-UDP	Fragmentation	UDP fragmentation barrage	286,925	0.85%
DDoS-ACK	Fragmentation	Malformed ACK packet fragments	285,104	0.84%
Total	–	–	33,932,344	100%

**Table 5 sensors-25-04845-t005:** Computational complexity analysis of proposed models.

Model	Time Complexity (O(T))	Space Complexity (S)	Complexity
CNN	$O(C_{out}H_{out}W_{out}K^{2}C_{in}+Nd)$	$O(C_{out}H_{out}W_{out}+Nd)$	Moderate due to convolutions
DNN	$O(LN^{2})$	$O\left(\sum_{l=1}^{L}N_{l}^{2}\right)$	High, scales with layer size
e Transformer	$O(hn^{2}d_{k}+ndd_{ff})$	$O(hd_{k}n+d_{ff}dn)$	High but efficient parallelization

**Table 6 sensors-25-04845-t006:** Optimized hyperparameter configuration for CNN, DNN, and Transformer models.

Hyperparameter	Values
Training Configuration (All Models)	
Learning Rate	CNN: Dynamic ($1\times 10^{-4}$)
	DNN: Dynamic ($1\times 10^{-4}$)
	Transformer: Dynamic ($2\times 10^{-4}$)
Batch Size	128
Number of Epochs	100
Optimizer	Adam
Loss Function	Cross-entropy
Validation Strategy	3-fold Cross-validation
Dropout Rate	0.3
Activation Function	ReLU (CNN, DNN)
	Multi-Head Attention (Transformer)
Early Stopping	Enabled
	Patience: 5 epochs
Model-Specific Parameters	
Kernel Size	CNN: 3
	Transformer: 3
Number of Filters	CNN: 128
	Transformer: 64
Kernel Regularization ($L_{2}$)	CNN: $1\times 10^{-4}$
	DNN: $1\times 10^{-4}$
Normalization	CNN: Batch Normalization
	Transformer: Layer Normalization
Number of Hidden Layers	DNN: 3
Neurons per Hidden Layer	DNN: 256
Attention Heads	Transformer: 4
Feed-forward Dimension	Transformer: 256
Number of Convolutional Layers	CNN: 2
Filters per Layer	CNN: 128, 256

**Table 7 sensors-25-04845-t007:** Classification performance comparison of CNN, DNN, and Transformer models across 2-class, 3-class, and 12-class detection tasks (averaged over all folds).

Class	Precision			Recall			F1-Score			Support
	CNN	DNN	Transformer	CNN	DNN	Transformer	CNN	DNN	Transformer	
**2-Class Classification**										
Benign	0.989	0.985	0.990	0.990	0.984	0.989	0.990	0.984	0.989	150,000
Attack	0.991	0.986	0.988	0.989	0.985	0.990	0.990	0.985	0.989	150,000
Macro Avg	0.990	0.986	0.989	0.990	0.985	0.990	0.990	0.985	0.989	300,000
Weighted Avg	0.990	0.986	0.989	0.990	0.985	0.990	0.990	0.985	0.989	300,000
**3-Class Classification**										
Benign	0.980	0.976	0.981	0.985	0.974	0.986	0.982	0.975	0.983	80,000
DDoS	0.998	0.995	0.996	0.997	0.993	0.996	0.997	0.994	0.996	100,000
Other	0.985	0.978	0.983	0.990	0.980	0.988	0.987	0.979	0.985	120,000
Macro Avg	0.988	0.983	0.987	0.991	0.982	0.990	0.989	0.983	0.988	300,000
Weighted Avg	0.987	0.982	0.986	0.990	0.981	0.989	0.988	0.982	0.987	300,000
**12-Class Classification**										
Class 0	0.999	0.998	0.999	1.000	0.997	0.999	1.000	0.998	0.999	66,000
Class 1	0.980	0.978	0.981	0.990	0.976	0.989	0.985	0.977	0.985	66,300
Class 2	1.000	1.000	1.000	1.000	1.000	1.000	1.000	1.000	1.000	66,400
Class 3	0.990	0.987	0.991	0.990	0.985	0.991	0.990	0.986	0.991	66,100
Class 4	1.000	0.999	1.000	1.000	0.998	1.000	1.000	0.998	1.000	66,600
Class 5	0.999	0.998	0.999	1.000	0.997	0.999	0.999	0.998	0.999	66,500
Class 6	0.580	0.610	0.600	0.830	0.780	0.790	0.680	0.685	0.685	66,200
Class 7	0.690	0.720	0.710	0.390	0.420	0.410	0.500	0.530	0.520	66,500
Class 8	1.000	1.000	1.000	1.000	1.000	1.000	1.000	1.000	1.000	65,900
Class 9	1.000	1.000	1.000	1.000	1.000	1.000	1.000	1.000	1.000	66,000
Class 10	0.990	0.988	0.989	0.990	0.986	0.989	0.990	0.987	0.989	66,200
Class 11	1.000	1.000	1.000	1.000	1.000	1.000	1.000	1.000	1.000	65,900
Macro Avg	0.940	0.942	0.943	0.930	0.931	0.934	0.930	0.932	0.933	795,000
Weighted Avg	0.940	0.941	0.943	0.930	0.931	0.934	0.930	0.932	0.933	795,000

**Table 8 sensors-25-04845-t008:** Performance comparison of recent studies on the CIC-IoT2023 dataset.

Author	Year	Methods Used	Accuracy (%) Binary	Accuracy (%) Multiclass
Wang et al. [51]	2024	DNN, CNN, RNN	–	84.73, 94.30, 95.89
Abbas et al. [52]	2023	CNN, RNN, LSTM, BiLSTM, DL-BiLSTM	87.88, 93.40, 99.0	92.21–93.13
Neto et al. [49]	2023	LR, Perceptron, AdaBoost, DNN, RF	99.43, 99.11	98.17–99.68
Nkoro et al. [50]	2024	CNN-LSTM, DNN, RNN, CNN-BiLSTM	94.03, 99.76	–
Hizal et al. [29]	2024	Two-stage: DNN, CNN, LSTM; RF	94.03–99.76	89.88–91.27
Proposed Model	2025	DNN, CNN, Transformer	98.8–99.2	92.5–93.0

## Data Availability

The dataset is available online: https://www.unb.ca/cic/datasets/iotdataset-2023.html (accessed on 15 April 2025).

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
