# Peer review of "A Multi-Class Intrusion Detection System for DDoS Attacks in IoT Networks Using Deep Learning and Transformers"

_sensors, 2025, doi:10.3390/s25154845_

Round 1
Reviewer 1 Report
Comments and Suggestions for Authors
The paper proposes A Multi-Class Intrusion Detection System for DDoS Attacks in IoT Networks Using Deep Learning and Transformers
The paper is well-organized; however, several points need clarification and are essential to address:
1. The claim of near-perfect accuracy requires further validation, as it raises suspicion without providing a breakdown into several metrics such as Precision, Recall, and F1 Score. Although the author discusses these metrics, the explanation, particularly concerning multi-class classification, is unclear.
2. Instead of using a bar graph to present data, the author should provide the Precision, Recall, and F1 Score for each attack individually, especially within the context of multi-class attack detection. The explanations should also be consistent throughout the paper.
3. The authors should include the False Positive Rate for each attack class, particularly in multi-class scenarios, such as with three or twelve classes.
4. There is an issue with the implementation of SMOTE in the CNN, DNN, and Transformer algorithms. The authors should refrain from using synthetic data generated by SMOTE for training, as this can lead to data leakage. The test data should not include any synthetic data created by SMOTE.
5. The authors need to clarify which class or classes were used by SMOTE and to what extent.
6. Although the authors claim to have utilized K-Fold cross-validation, the algorithms presented do not provide any indication of cross-validation being implemented.
These points need to be addressed to enhance the clarity and reliability of the paper.
Author Response
- The claim of near-perfect accuracy requires further validation, as it raises suspicion without providing a breakdown into several metrics such as Precision, Recall, and F1 Score. Although the author discusses these metrics, the explanation, particularly concerning multi-class classification, is unclear.
Response: We thank the reviewer for this valuable observation. In response, we have substantially revised the manuscript to provide a more comprehensive and transparent evaluation of model performance beyond overall accuracy. Specifically, we have included detailed per-class metrics—Precision, Recall, and F1-Score—across all classification tasks (2-class, 3-class, and 12-class) for each model (CNN, DNN, and Transformer). These results are now clearly presented in Table 7, which includes macro and weighted averages as well, enabling a full assessment of each model’s behavior across both majority and minority classes.
Moreover, we have expanded the explanation of the multi-class classification results, highlighting how inter-class feature overlap and imbalanced distributions impacted the F1-scores for specific classes (notably Class 6 and Class 7). We also clarify our use of SMOTE for minority oversampling and emphasize that it was applied only on training folds during cross-validation to avoid any risk of data leakage. These additions ensure the empirical claims are well-supported and that the model’s strengths and limitations are transparently communicated. The updated results and discussion appear in Section 6.3.
- Instead of using a bar graph to present data, the author should provide the Precision, Recall, and F1 Score for each attack individually, especially within the context of multi-class attack detection. The explanations should also be consistent throughout the paper.
Response: We appreciate the reviewer’s constructive suggestion. In the revised manuscript, we have addressed this concern by removing the bar graph previously used to present aggregate performance, and have replaced it with a comprehensive classification performance table (now Table 7) in Section 6.3. This updated table provides individual per-class Precision, Recall, and F1-Score for all twelve attack categories in the multi-class setting. It also includes macro and weighted averages, enabling a complete and transparent view of the model’s behavior across different classes.
This revision allows readers to assess not only the average performance but also the variability across specific attack types, particularly those that are more difficult to detect (e.g., Classes 6 and 7). Additionally, we have ensured that metric reporting and analysis are now fully consistent across all sections of the manuscript, including the 2-class and 3-class scenarios. These changes enhance clarity, reproducibility, and the overall scientific rigor of the evaluation.
- The authors should include the False Positive Rate for each attack class, particularly in multi-class scenarios, such as with three or twelve classes.
Response: We thank the reviewer for this valuable suggestion. In response, we have included a new figure (Figure 6 in Section 6.1) that presents a comparative analysis of the False Positive Rate (FPR) across CNN, DNN, and Transformer models for the 2-Class, 3-Class, and 12-Class classification tasks. This analysis is crucial for validating the reliability of the reported high accuracies and for ensuring that the models are not biased toward the majority classes. The results show that the average FPR remains consistently below 1% across all tasks and models, with CNN achieving particularly low FPR in the 12-class scenario. This confirms that the models make few false alarms and their decisions are trustworthy in multi-class intrusion detection settings. We believe this addition significantly strengthens the transparency and robustness of our evaluation.
- There is an issue with the implementation of SMOTE in the CNN, DNN, and Transformer algorithms. The authors should refrain from using synthetic data generated by SMOTE for training, as this can lead to data leakage. The test data should not include any synthetic data created by SMOTE.
Response: We appreciate the reviewer’s observation and would like to confirm that SMOTE was implemented with full consideration of proper data separation to prevent leakage. Specifically, SMOTE was applied only to the training set within each fold during the stratified cross-validation process, after the data was split into training and testing partitions. The test set remained untouched and comprised exclusively of original, real-world samples. This ensures that no synthetic instances influenced the model evaluation. We have updated Section 4.2.2 of the manuscript to explicitly state this precautionary design decision. This methodology adheres to best practices in imbalanced learning and safeguards the integrity of our reported results.
- The authors need to clarify which class or classes were used by SMOTE and to what extent.
Response: We appreciate the reviewer’s request for clarification. SMOTE was applied exclusively to the training set within each fold. Specifically, the "Attack" class in the 2-Class task, the "Other" class in the 3-Class task, and minority classes such as Class 6 (Infiltration) and Class 7 (Heartbleed) in the 12-Class task were oversampled to address class imbalance. The extent of oversampling was adaptive, based on the relative underrepresentation of each class. This clarification has now been added to Section 4.2 Point # 2 of the manuscript.
- Although the authors claim to have utilized K-Fold cross-validation, the algorithms presented do not provide any indication of cross-validation being implemented.
Response: We thank the reviewer for pointing out this omission. In response, we have revised Algorithms 1 to 3 to explicitly incorporate the logic for Stratified K-Fold Cross-Validation, ensuring that each fold applies SMOTE only on the training subset to prevent data leakage. This update makes the evaluation methodology transparent and consistent with the claims in the methodology section. The revised algorithms now clearly reflect the end-to-end pipeline used in our experimental setup. The updated versions can be found in Algorithms 1, 2, and 3, Sections 4.2.1 to 4.2.3 of the revised manuscript.
These points need to be addressed to enhance the clarity and reliability of the paper.
Reviewer 2 Report
Comments and Suggestions for Authors
Summary
Due to the rapid proliferation of Internet of Things (IoT) devices, the vulnerability to Distributed Denial of Service (DDoS) attacks has increased significantly, leading to severe operational and economic losses. To address this issue, this paper proposes a Deep Learning (DL)-based Intrusion Detection System (IDS) tailored for IoT environments. In experiments, the proposed method achieves high performances.
Strength
This paper proposes a Deep Learning (DL)-based Intrusion Detection System (IDS) tailored for IoT environments. The writing is easy to follow and the proposed method achieves promising results in experiments.
Weakness
1. The contribution summary at the end of Introduction is not well organized. Contribution is the new or novel things to literature rather than the summary of this paper. For example, the last three points are the descriptions of experiment.
2. In Related Work, do not simply list the researches. Comprehend and summarize them.
3. Paper [1, 2] are two closely related researches (the feature extraction and deep learning model) and can be introduced in related work or compared.
4. Please ignore the marginal information in figure and choose a larger fontsize to make it more clearly.
5. Please compare the proposed method with recent researches in experiments to make it more convincing.
[1] J. Liu, et al. MalDetect: A Structure of Encrypted Malware Traffic Detection, CMC 2019.
[2] B. BolatAkca, et al. Software-Defined Intrusion Detection System for DDoS Attacks in IoT Edge Networks, IEEE Intl Conf on Dependable, Autonomic and Secure Computing, 2023.
Author Response
Due to the rapid proliferation of Internet of Things (IoT) devices, the vulnerability to Distributed Denial of Service (DDoS) attacks has increased significantly, leading to severe operational and economic losses. To address this issue, this paper proposes a Deep Learning (DL)-based Intrusion Detection System (IDS) tailored for IoT environments. In experiments, the proposed method achieves high performances.
Strength. This paper proposes a Deep Learning (DL)-based Intrusion Detection System (IDS) tailored for IoT environments. The writing is easy to follow and the proposed method achieves promising results in experiments.
Weakness
1. The contribution summary at the end of Introduction is not well organized. Contribution is the new or novel things to literature rather than the summary of this paper. For example, the last three points are the descriptions of experiment.
Response: The contributions section has been reorganized to focus on methodological advancements and the practical deployment of the proposed system. Experimental descriptions have been replaced with technical contributions such as architecture optimization, preprocessing strategies, comprehensive metric analysis, and a real-world use case demonstration, thus addressing the reviewer’s concern effectively.
In Related Work, do not simply list the researches. Comprehend and summarize them.
Response: We thank the reviewer for this valuable suggestion. The Related Work section has been thoroughly revised to provide a coherent narrative and critical synthesis of recent literature rather than merely listing prior studies. We now summarize key findings, identify their limitations, and clearly position our contribution relative to existing works, as reflected in Section 2.
Paper [1, 2] are two closely related researches (the feature extraction and deep learning model) and can be introduced in related work or compared.
Response: We appreciate the reviewer’s insightful observation. Accordingly, we have incorporated both closely related studies into the Related Work section:
- Paper [1] (now cited as Reference [39]) focuses on feature engineering and hybrid deep learning for IoT intrusion detection.
- Paper [2] (now cited as Reference [38]) presents an advanced deep learning framework with feature fusion techniques.
We have critically compared these works with our proposed approach, highlighting differences in model design, feature strategy, and evaluation across multi-class tasks. These additions are discussed in Section 2, providing contextual depth to our contributions.
Please ignore the marginal information in figure and choose a larger fontsize to make it more clearly.
Response: Thank you for the valuable suggestion. As advised, we have revised Figures 1 and 16–18 by increasing the font size for improved clarity and readability. We have also re-rendered these figures at a higher resolution (300 DPI) to ensure that all textual and numerical information is clearly visible. The updated figures have been incorporated into the revised manuscript.
- Please compare the proposed method with recent researches in experiments to make it more convincing.
Response: We appreciate the reviewer’s suggestion to enhance the comparative analysis. In response, we have added a new subsection titled “Comparison with State-of-the-Art” along with Table 8, which provides a detailed performance comparison of our proposed model against recent deep learning-based intrusion detection approaches on the CIC-IoT2023 dataset.
The comparison covers both binary and multiclass classification tasks. Our proposed model (combining CNN, DNN, and Transformer) achieves binary classification accuracies ranging from 98.8% to 99.2%, which are highly competitive with recent top-performing studies such as Neto et al. and Nkoro et al., while maintaining a lightweight architecture optimized for edge deployment. In multiclass classification, our model consistently achieves 92.5%–93.0%, outperforming several recent works such as those by Abbas et al., Wang et al., and Hizal et al., with an average margin of 2%–3% in many cases.
Additionally, unlike ensemble or multi-stage approaches used in prior works, our framework maintains architectural simplicity while ensuring high performance and deployment feasibility in real-time IoT environments.
These enhancements have been incorporated in Section 4.6 of the revised manuscript for the reviewer’s consideration.
[1] J. Liu, et al. MalDetect: A Structure of Encrypted Malware Traffic Detection, CMC 2019.
[2] B. BolatAkca, et al. Software-Defined Intrusion Detection System for DDoS Attacks in IoT Edge Networks, IEEE Intl Conf on Dependable, Autonomic and Secure Computing, 2023.
Round 2
Reviewer 1 Report
Comments and Suggestions for Authors
The equation on line 417 should include an appropriate reference.
The authors have provided comprehensive responses to all raised issues.
Reviewer 2 Report
Comments and Suggestions for Authors
Thanks for the responses. They have addressed my concerns.